# A Review on the Phytochemistry, Medicinal Properties and Pharmacological Activities of 15 Selected Myanmar Medicinal Plants

**DOI:** 10.3390/molecules24020293

**Published:** 2019-01-15

**Authors:** Mya Mu Aye, Hnin Thanda Aung, Myint Myint Sein, Chabaco Armijos

**Affiliations:** 1Department of Chemistry, Mandalay University, Mandalay 100103, Myanmar; myamuaye11@gmail.com (M.M.A.); myintmyintkyawsein@gmail.com (M.M.S.); 2Department of Chemistry, Kalay University, Kalay 03044, Sagaing Region, Myanmar; hninthandaaung07@gmail.com; 3Departamento de Química y Ciencias Exactas, Universidad Técnica Particular de Loja (UTPL), San Cayetano Alto, Loja 11 01 608, Ecuador

**Keywords:** Myanmar, medicinal plants, phytochemistry, ethnopharmacology, ethnomedicine

## Abstract

Medicinal plants are a reservoir of biologically active compounds with therapeutic properties that over time have been reported and used by diverse groups of people for treatment of various diseases. This review covers 15 selected medicinal plants distributed in Myanmar, including *Dalbergia cultrata*, *Eriosema chinense*, *Erythrina suberosa*, *Millettia pendula*, *Sesbania grandiflora*, *Tadehagi triquetrum*, *Andrographis echioides*, *Barleria cristata*, *Justicia gendarussa*, *Premna integrifolia*, *Vitex trifolia*, *Acacia pennata*, *Cassia auriculata*, *Croton oblongifolius* and *Glycomis pentaphylla*. Investigation of the phytochemical constituents, biological and pharmacological activities of the selected medicinal plants is reported. This study aims at providing a collection of publications on the species of selected medicinal plants in Myanmar along with a critical review of the literature data. As a country, Myanmar appears to be a source of traditional drugs that have not yet been scientifically investigated. This review will be support for further investigations on the pharmacological activity of medicinal plant species in Myanmar.

## 1. Introduction

Natural products, especially those derived from plants, have been used to help mankind sustain human health since the dawn of medicine. Traditional medicine has been in existence since time immemorial and has been well accepted and utilized by the people throughout history. Since ancient times, plants have been an exemplary source of medicines. Plant-derived medicinal products have attracted the attention of scientists around the world for many years due to their minimum side effects and positive effects on human health.

In the pharmaceutical landscape, plants with a long history of use in ethnomedicine can be a rich source of substances for the treatment of various ailments and infectious diseases. Medicinal plants are considered a repository of numerous types of bioactive compounds possessing varied therapeutic properties. The vast array of therapeutic effects associated with medicinal plants includes anti-inflammatory, antiviral, antitumor, antimalarial, and analgesic properties.

According to the World Health Organization (WHO), a variety of drugs are obtained from different medicinal plants and about 80% of the world’s developing population depends on traditional medicines for their primary health care needs [1]. Myanmar (Figure 1) has abundant plant resources and Myanmar peoples have used their own traditional medicines to maintain their health and treat various ailments, including malaria, diarrhea and fever over millennia of history [2].

We previously reported [3] the traditional uses, chemical constituents, the published the biological and pharmacological evidences of 25 medicinal plants from Kachin State, in Northern Myanmar. In Kachin State, the climatic conditions vary considerably, from warm to humid in the lowlands, to extremely cold in the highlands, and the mountains in the far north are snowcapped. The geographical and climate conditions of the central region in Myanmar are quite different from those of Kachin State. That region has a generally harsh climate and is extremely dry. Because of the geographical and climate differences the chemical constituents of the plants from that region have different medicinal uses, biological and pharmacological activities. Therefore, in this review, it is out intention to provide a complete account of the phytochemistry, medicinal and pharmacological uses of a few plants which grow in the central region of Myanmar, which is the tropical area of Myanmar. The traditional healers in central Myanmar use a lot of species of plants that serve as regular sources of medicine and there is a need to preserve the traditional knowledge on the uses of these medicinal plants.

A total of 15 medicinal plants, which belong to seven families and 15 genera were recorded in this review. The results are summarized in Table 1, which provides the following information for each species: scientific name, botanical family and local name. The most represented families are the Fabaceae with six species, followed by Acanthaceae (three species), Verbenaceae (two species), and Mimosaceae, Casealpiniaceae, Euphorbiaceae and Rutaceae were represented with one species each. The primary objective of this study is to present a database of the indigenous knowledge on medicinal plants used for various diseases among the local traditional healers of Myanmar, to describe the chemical constituents and to summarize the published biological and pharmacological evidence. The focus of this review is to provide information on the phytochemical constituents reported in the literature and published biological and pharmaceutical activities of fifteen medicinal plants from Myanmar.

Until now, few sources of information are available for some medicinal plants in the literature, and the sources of information in the literature make evident that additional new investigations and scientific reports are necessary.

## 2. Phytochemistry, Medicinal Properties and Ethnopharmacology of the Selected Myanmar Medicinal Plants

### 2.1. Dalbergia culrata Grah. (DC)

*D. cultrata*, locally known as Yin-daik, is widely distributed in Myanmar [4]. It is a moderate-sized deciduous tree indigenous to the forests of Burma that belongs to the family Fabaceae (or Leguminosae) [5]. This plant genus is mainly distributed in tropical and subtropical regions, such as South Africa and India, Laos, Myanmar, Thailand and Vietnam in Asia [6,7].

The plant family is known to contain isoflavonoids and neoflavonoids [8]. The chemical constituents from the heartwood of this plant includes 4-arylcoumarin derivatives; (*S*)-4-methoxy dalbergione (**1**) and stevenin (**2**), a *p*-hydroquinone derivative, dalbergin (**3**), and diphenylpropene derivatives 3,3-diphenylprop-1-ene (**4**) and 3-(5-acetoxy-2,4-dimethoxyphenyl)-3-(3′-acetoxyphenyl) prop-1-ene (**5**) [9]. Ito isolated cinnamoylphenol type compounds, dalberatin A (**6**) and dalberatin B (**7**) (Figure 2) from the stem bark of *D. cultrate* which showed remarkably potent tumor-promotion inhibitory activity in vitro with the IC_50_ value of 212 and 303 (mol ratio/32 pmol TPA), suggesting that they might be valuable as potential cancer chemopreventive agents [10].

### 2.2. Eriosema chinense Vogel.

*E. chinense* (family Fabaceae) is locally known in Myanmar as pike-san-gale. It is a small plant and two members of its genus are found in Myanmar. In Myanmar traditional medicine, a decoction of the seeds is used for scrofula, diarrhoea, to treat leucorrhoea and menstrual derangements. In 2009, Sutthivaiyakit’s group isolated eight prenylated flavonoids, khonklonginols A–H (**8**–**15**), five flavonoids, lupinifolinol (**16**), dehydrolupinifolinol (**17**), flemichin D (**18**), eriosemaone A (**19**), lupinifolin (**20**), and one lignin, yangambin, from the hexane and dichloromethane extracts of the roots of *E. chinense* [11] (Figure 3).

From the hexane, dichloromethane and methanol extract of the roots of *E. chinense*, seven flavonoids, *trans*-*p*-coumaric acid ester together with another twelve compounds were identified as 3-epilupinifolinol (**21**), 3-epikhonklonginol C (**22**), 2′-hydroxylupinifolinol (**23**), 2,5,2,4-tetrahydroxy-6″,6″-dimethylpyrano(2″,3″:7,6)-8-(3‴,3‴-dimethylallyl) flavone (**24**), (2*R*,3*R*,2‴*R*)-3,5,2‴-trihydroxy-4-methoxy-6″,6″-dimethylpyrano(2″,3″:7,6)-8-(3‴-methylbut-3‴-enyl) flavanone (**25**), 2‴,3‴-epoxy-khonklonginol A (**26**), 6,7-dimethoxy-5,2′,4′-trihydroxyisoflavone (**27**), octaeicosanyl-*trans*-*p*-coumarate, khonklonginol A (**8**), lupinifolinol (**16**), flemichin D (**18**), 7-*O*-methyltectorigenin, tectorigenin (**28**), genistein (**29**), kaempferol (**30**), 2′,4′,5,7-tetrahydroxy-6-methoxyisoflavone, kaempferol-7-*O*-*β*-d-glucopyranoside (**31**), genistein-7-*O*-*β*-d-glucopyranoside, genistin, astragalin (**32**) and cajanol [12].

Prasad evaluated the antidiarrhoeal activity of the roots of ethanol extract, its fractions (chloroform, ethyl acetate and hexane) and lupinifolin isolated from the chloroform fraction of the roots. Lupinifolin showed moderate antidiarrhoel activity compared to the bioactive chloroform fraction [13].

A prenylated flavanone, eriosematin E (**33**) was isolated from the roots of *E. chinense* and significant antidiarrhoel activity results were observed, with the maximum effective dose of eriosematin E (10 mg/kg p.o.) in inhibiting peristaltic index (small intestinal transit) and reducing intestinal fluid volume of castor oil induced and PGE 2 induced enteropooling models. In addition, eriosematin E restored all the alterations in biochemical parameters such as nitric oxide, protein, DNA, superoxide dismutase, catalase and lipid peroxidation and also significantly recovered Na^+^ and K^+^ loss from body [14]. The cytotoxic inhibitory activity of khonklonginol A (**8**), khonklonginol B (**9**) and lupinifolinol (**16**) showed IC_50_ values of 3.1, 3.8 and 1.73 μg/mL against cell lung NCI-H187 cell line. Dehydrolupinifolinol (**17**), flemichin D (**18**), eriosemaone A (**19**) and lupinifolin (**20**) were showed the antimycobacterial against *Mycobacterium tuberculosis* H37Ra with MIC values, in turn, 25, 12.5, 12.5 and 12.5 μg/mL [11]. Khonklonginol A (**8**), lupinifolinol (**16**), flemichin D (**18**), 3-epi-lupinifolinal (**21**), 2′-hydoxylupinifolinol (**23**), 3,5,2,4-tetrahydroxy-6″,6″-dimethylpyrano (2″,3″:7,6)-8-(3‴,3‴-dimethylallyl) flavone (**24**), tectorigenin (**28**), genistein (**29**), kaempferol (**30**), kaempferol-7-*O*-β-d-glycopyranoside (**31**) and astragalin (**32**) showed antimicrobial and antioxidant activity [12]. Among these compounds, khonklonginol A showed the most effectively activity against *Candida albicans*, gram negative bacteria, *Escherichia coli*, *Klebsiella pneumoniae* and *Pseudomonas aeruginosa* and positive bacteria, *Bacillus cereus*, *Enterococcus faecalis*, *Staphylococcus aureus*, MRSA *S. aureus*, *S. epidermidis*, *S. agalactiae* and *S. pyrogenes*. A detailed description of the antimicrobial and antioxidant activities of these compounds is given in Table 2. The antidiarrhoel activity of the various extracts of the roots of *E. chinense* has been reported [13].

### 2.3. Erythrina suberosa Roxb.

*E. suberosa* locally known as ka-thit and widely distributed in Myanmar. It belongs to the family Fabaceae, and is an ornamental tall tree also found occasionally on hills and moist slopes of India, Pakistan, Nepal, Bhutan, Burma, Thailand and Vietnam [15]. In traditional Myanmar medicine, the bark is used for dysentery, while the leaves are reported to be used in urinary diseases and inflammation. In India, *E. suberosa* has been used as a very important medicinal plant for the treatment of various ailments [16]. The bark is reported to be used in preparation of medicines for dysentery, as an astringent and febrifuge, and used for anorexia, liver troubles, helminthic manifestations, inflammations, intestinal worms, obesity and as a snake-bite antidote [17].

The species of this genus are famous for different classes of compounds like alkaloids, flavonoids and terpenes. One prenylated flavanone 5,7,3′,4′-tetrahydroxy-2′-(3″-methyl but-2″-enyl)-flavanone (**34**) (Figure 4) isolated from the ethanol extract of the roots of this plant [18]. According to the Kumar group’s report, the stem bark of *E. suberosa* contains α-hydroxyerysotrine (**35**), 4′-methoxy licoflavanone (**36**), alpinumisoflavone (**37**) and wighteone (**38**) [19].

The seed powder of *E. suberosa* contains erythraline (**39**), erysodine (**40**), erysotrine (**41**) and hypaphorine [20]. Serrano’s group isolated two erythrinian alkaloids, erysodine and erysotrine, from the ethanolic extract of the flowers of *E. suberosa* [15]. The dichloromethane extract of the wood of *E. suberosa* is reported to contain erysubin A (**42**), erysubin B (**43**), erythrinin C (**44**), alpinumisoflavone, a wighteone metabolite, wighteone and laburnetin [21]. Dichloromethane extract of the roots of this plant have been reported to contain petrocarpans, erysubin C (**45**), erysubin D (**46**), erysubin E (**47**), erysubin F (**48**), orientanol D, bidwillon B, 2-(γ,γ-dimethylallyl)-6a-hydroxyphaseollidin, phaseollin, erythrabyssin II, cristacarpin, wighteone, bidwillon A, alpinumisoflavone and erystagallin A [22].

Serrano observed that acute p.o. treatment with the isolated alkaloids; erysodine (**40**) and erysotrine (**41**) produced anxiolytic effects in the elevated plus-maze and the light-dark transition model [15]. The anticancer potential and cytotoxic effect of some compounds from *E. suberosa* has been reported scientifically. Thus, Kumar studied the cytotoxic effects of 4′-methoxylicoflavanone (**36**) and alpinumiso-flavone (**37**) from *E. suberosa* on apoptosis in human leukemia HL-60 cells and their potential to induce cancer cell death [19]. The ethanolic extract of the leaves of *E. suberosa* showed antitumor activity [23]. In addition, the ethanolic extract of the stem bark of *E. suberosa* induced apoptosis in human promyelocytic leukemia HL-60 cells [24]. It seems that *E. suberosa* has the potential for being selected for future anticancer therapeutics.

### 2.4. Millettia pendula BENTH.

*M. pendula* is locally called thin-win and is widely distributed in Myanmar. *M. pendula* also belongs to the family Fabaceae. Its timber is produced in Myanmar and Thailand [25]. In Myanmar traditional medicine, the root of *M. pendula* is applied for skin diseases, allergy, urinary diseases and inflammation, based on folklore medicine. Hayashi found the pterocarpan, maackiain and isoflavans, pendulone and claussenquinone in the heartwood of *M. pendula* and confirmed their identity and purification through NMR [26].

In 2006, three isoflavans: millettilone A (**49**), 3R-claussequinone (**50**) and pendulone (**51**), a 2-arylbenzofuran—millettilone B (**52**)—three pterocarpans: secundiflorol I (**53**), 3,8-dihydroxy-9-methoxypterocarpan (**54**) and 3,10-dihydroxy-7,9-dimethoxypterocarpan (**55**), and one isoflavone, formononetin (**56**), were isolated from a methanol extract of the powdered *M. pendula* timber [25] (Figure 5).

Takahashi reported the vitro leishmanicidal activity of the methanolic timber extract of *M. pendula* [25]. Among the constituents of *M. pendula* pendulone (**51**), an isoflavan, showed significant leishmanicidal activity with an IC_50_ value of 0.07 μg/mL, while millettilone A (**49**, IC_50_ 9.3 μg/mL), 3*R*-claussequinone (**50**, IC_50_ 1.2 μg/mL) and 3,8-dihydroxy-9-methoxypterocarpan (**54**, IC_50_ 2.9 μg/mL) exhibited moderate activity and secundiflorol I (**53**, IC_50_ 86 μg/mL) and 3,10-dihydroxy-7,9-dimethoxypterocarpan (**55**, IC_50_ 77 μg/mL) displayed weak leishmanicidal activity in vitro [25]. There are few chemical constituents and biological studies of this plant reported in the scientific literature, therefore, biologically active new chemical components from this plant might yet be found.

### 2.5. Sesbania grandiflora (L.) Poir.

*S. grandiflora* is a small, erect, fast-growing, and sparsely branched cultivated tree belonging to the family Fabaceae. It’s locally known as paukpan-phyu in Myanmar. In folk medicine, the juice of the leaves and flowers is used as a popular remedy for catarrh, and headache. The bark of *S. grandiflora* is very astringent and the bitter bark is considered a tonic. Decoctions of leaves and flowers is used to treat leucorrhoea and vomiting of blood. The bark of *S. grandiflora* is used as an astringent and treatment of small pox, ulcers in the mouth and the alimentary canal, infant stomach disorders and scabies. The juice of the leaves of *S. grandiflora* has been reportedly used in the treatment of bronchitis, cough, vomiting, wounds ulcers, diarrhea, and dysentery. The flowers have reported antimicrobial activity. Powdered roots of this plant are mixed in water and applied externally as a poultice or rub for rheumatic swelling [27,28].

According to qualitative analysis, the methanol extract of the leaves of *S. grandiflora* is reported to contain alkaloids, glycosides, steroids, terpenoids and tannins. 3,4,5-Trimethoxyphenol, erucic acid, 2-furancarboxaldehyde, vitamin E acetate, 4-methyloxazole, palmitic acid, 9-hexadecenol, and dioctyl ester are the major compounds in the leaves of *S. grandiflora* by GC-MS analysis [29]. Three isoflavonoids—isovestitol (**57**), medicarpin (**58**) and sativan (**59**)—and one lupane type triterpene, betulinic acid (**60**, Figure 6) were isolated from the methanol extract of the roots of *S. grandiflora* [30]. High contents of quercetin, myricetin and kaempferol were identified in a methanolic extract of the leaves [31] and a novel protein fraction was isolated from the fresh flowers, which displayed chemopreventive effects [32]. Isovestitol (**57**), medicarpin (**58**) and sativan (**59**) all displayed significant antitubercular activity, with MIC values of 50 μg/mL each, but betulinic acid (**60**) showed weak activity against *Mycobacterium tuberculosis* H37Rv with a MIC value of 100 μg/mL in vitro. The crude extract of the root of *S. grandiflora* exhibited a moderate antitubercular activity [27].

Remesh has reported that *S. grandiflora* has protective effects against cigarette smoke-induced oxidative damage in the liver and kidney of rats and protected the lungs from oxidative damage through its antioxidant potential [33,34]. Doddola reported the antiurolithiatic and antioxidant activities of the leaf juice of *S. grandiflora* [35]. The ethanol extract of the leaves and flowers of *S. grandiflora* exhibited anticancer activity against an Ehrlich ascites carcinoma tumor model [36]. The leaves of *S. grandiflora* showed a significant protection against erythromycin estolate-induced toxicity [27], antimicrobial [37], anticonvulsant and anxiolytic activity [38]. The alcoholic extract of *S. grandiflora* flowers at 250 and 500 mg/kg exhibited significant antidiabetic activity (*p* < 0.01) in alloxan- induced diabetic rats [39]. The *S. grandiflora* flowers displayed significant hepatoprotective [40] and antimicrobial activity [28]. Silver nanoparticles synthesized using the leaf extract of this plant showed not only excellent antibacterial activity against clinically isolated multi-drug resistant human pathogens [41], but also antifungal activity against some selected microbial pathogens [42].

### 2.6. Tadehagi triquetrum (L.) H. Ohashi.

*T. triquetrum* (synonyms: *Desmodium triquetrum* (L.) DC.) belongs to the family Fabaceae, which is widely distributed throughout the tropical, subtropical and Pacific regions of the world. This shrub is commonly known as lauk-thay or shwe-gu-than-hlet in Myanmar and is distributed in the regions of Chin, Kachin, Kayin, Mandalay, Sagaing, Shan and Yangon [4] and the southern area of Yunan Province in China [43]. An infusion of the roots is taken to treat kidney complaints, and an infusion of the leaves is drunk for stomach discomfort in Myanmar. A daily glass of a decoction of the roots of this plant is considered beneficial for chronic coughs and tuberculosis.

In the whole plant of *T. triquetrum*, Xiang et al., in 2005 and Zhang et al., in 2011 reported prenylated isoflavones—triquetrumone A (**61**), triquetrumone B (**62**), and triquetrumone C (**63**)—a prenylated biisoflavanone-(*R*)—triquetrumone D (**64**)—together with cyclokievitone, yukovanol, aromadendrin, kaempferol, astragalin, 2-*O*-methyl-l-chiro-inositol, galactitol, *p*-hydroxycinnamic acid, ursolic acid, betulinic acid, β-sitosterol, daucosterol, stigmasterol, stigmasta-5,22-dien-3-*O*-*β*-d-glucopyranoside, saccharose and docosanoic acid [43], and isoflavanones: triquetrumones E–H (**65**–**68**) [44]. From the aerial parts of *T. triquetrum*, one lignin—tadehaginosin (**69**)—along with 3,4-dihydro-4-(4′-hydroxyphenyl)-5,7-dihydroxycoumarin (**70**) [45], and ten phenylpropanoid glucosides-tadehaginosides A–J (**71**–**80**), along with tadehaginoside (**81**) were isolated [46] (Figure 7).

In Myanmar, local people use *T. triquetru* leaves to cover local fish pastes to prevent the development of fly larvae. A study in Myanmar proved that the aqueous extract of *T. triquetrum* has bactericidal activity on some pathogenic bacteria [47]. Moe et al. reported that the leaves of *T. triquetrum* were used by tuberculosis patients in Myanmar. The family members of patients were also familiar with this plant. In the central region of Myanmar, the herbs are available in the forest fringes but sometimes available in local markets of rural areas [48]. The extract of *T. triquetrum* and tadehagenoside proved to display hypoglycemic activity [49], with in vivo immunoprotection effects because of its antioxygenic properties [50], and anti-inflammatory activity [51]. Moreover, tadehaginosin (**69**) and 3,4-dihydro-4-(4′-hydroxyphenyl)-5,7-dihydroxycoumarin (**70**) from *T. triquetrum* showed hypoglycemic activity in vitro on HepG2 cells [45]. The phenylpropanoid glucosides-tadehaginosides C–J (**73**–**80**) and tadehaginoside (**81**) isolated from *T. triquetrum* showed the potential to be developed into antidiabetic compounds [46], providing the scientific evidence needed for developing new chemical candidates in treating diabetes. In Myanmar, there are more and more diabetes patients nowadays, and based on the literature review, the plant *T. triquetrum* could be a very useful and valuable plant for treating diabetes patients.

### 2.7. Andrographis echioides Nees.

*A. echioides* is an annual medicinal plant of Acanthaceae family and the plants are seen mostly in dry places, such as South India, Sri Lanka and South Asian countries [52]. In Myanmar, it is distributed in the central region—Mandalay and Sagaing—and known by the common name se-ga-gyi-hmawe-tu. There is no traditional knowledge of *A. echioides* use in Myanmar till now based on folk medicine, but in Indian traditional medicine, the leaf juice of *A. echioides* is used to cure fevers [53]. The leaves of *A. echioides* boiled with coconut oil is applied to decrease the loss and graying of hair [54].

From the whole plant of *A. echioides*, Jayaprakasam’s group characterized flavanone- dihydroechioidinin (**82**) (Figure 8) [55], flavones—echioidinin (**83**) [56], echioidin (**84**), skullcapflavone I 2′-*O*-methyl ether (**85**)—flavone glucosides—skullcapflavone I 2′-*O*-glucoside (**86**) [57,58], and echioidinin 5-*O*-β-d-glucoside (**87**) [59] and a chalcone glucoside, androechin (**88**). 2′-Oxygenated flavonoids and phenyl glycosides from the whole plant of *A. echioides* include androgechosides A–B (**89**,**90**), androechiosides A–B (**91**,**92**) together with 2′,6′-dihydroxy-acetophenone 2′-*O*-β-d-glucopyranoside (**93**), pinostrobin (**94**), andrographidine C (**95**), dihydroechioidinin, tectochrysin 5-glucoside (**96**), methyl salicylate glucoside, 7,8-dimethoxy-5-hydroxyflavone (**97**), 5,7,8-trimethoxyflavone (**98**), skullcapflavone I 2′-methyl ether, acetophenone-2-*O*-β-d-glucopyranoside, androechin, skullcapflavone I 2′-*O*-β-d-glucopyranoside, tectochrysin (**99**), negletein 6-*O*-β-d-glucopyranoside, andrographidine E, 4-hydroxy-3-methoxy-*trans*-cinnamic acid methyl ester, 4-hydroxybenzaldehyde, 4-hydroxy-*trans*-cinnamic acid methyl ester, *O*-coumaric acid, 2,6-dihydrobenzoic acid, 132-hydroxy-(132-*R*)-phaeophytin, (*E*)-phytyl epoxide, phytol, phytene 1,2-diol, (+)-dehydrovomifoliol, 3β-hydroxy-5α,6α-epoxy-7-megastigmen-9-one, β-sitosterol, β-sitosteryl-3-*O*-β-glucopyranoside, squalene, 1*H*-indole-3-carbaldehyde, and loliolide were reported by Shen’s group [58]. Moreover, a variety of chemical compounds isolated from the petroleum ether, ethyl acetate and methanol extracts of *A. echioides* were analysed by GC-MS [60].

A flavanone, dihydroechioidinin (**82**), and 5,7,8-trimethoxyflavone (**98**) isolated from *A. echioides* are reported to possess potent anti-inflammatory activity, with IC_50_ values of 37.6 ± 1.2 μM and 39.1 ± 1.3 μM, respectively, and to suppress NO inhibition activity [58].

Basu reported not only the hepatoprotective and antioxidant activity of methanolic extract of the aerial parts of *A. echioides* against acetaminophen-induced hepatotoxicity in rats, but also the anti-inflammatory, analgestic and antipyretic activities of the ether, chloroform and ethyl acetate extracts of this plant [61,62]. The chloroform extract of the leaves of *A. echioides* exhibited significant diuretic activity [63]. The ethanol extract from the leaves of *A. echioides* showed antiulcer activity [64] and antifungal activity in vitro [65]. Rajkumar et al., in 2012 reported that the synergistic larvicidal activity of the leaf extracts of *A. echioides* against larvae of *Aedes aegypti* L [66]. Thus, we can say that *A. echioides* is a good source of antioxidant and antimicrobial compounds. The ethanolic extract of *A. echioides* has notable antibacterial activity against both Gram positive and negative bacteria in vitro [67] and the hydroalcoholic extract of this plant showed antimicrobial activity [54]. The methanol extract of *A. echioides* displayed potent antioxidant activity IC_50_ values on the range of 1.14 ± 0.06 to 1.15 ± 0.45 mg/mL, as well as antibacterial activity (16 ± 1.527 mm) than the aqueous extract of this plant (20 ± 1.527 mm) [68]. While the phytochemical constituents and biological activities of the extracts of *A. echioides* have been reported, however, the pharmacological action of pure compounds from this plant has not been investigated yet, and therefore might represent an interesting research topic.

### 2.8. Barleria cristata L.

*B. cristata* or leik-tha-ywe-pya is a perennial shrub, belonging to the family Acanthaceae. It is found widely in subtropical Himalaya, Sikkim, Kashi Hills, and southern India at a height of 1350 meter [69]. The bitter juice of the leaves of *B. cristata* is used as a diaphoretic and expectorant for serious catarrhal infections in the central region of Myanmar. In addition, an infusion of the roots and leaves is applied to boils and sores to reduce swelling. *B. cristata* is traditionally used for the treatment of a variety of diseases including anemia, toothache and coughs in Indian traditional medicine. Roots and leaves are used to reduce swelling and inflammation [70].

Phytochemical screening by HPTLC analysis revealed that the presence of amino acids, carbohydrates, flavonoids, glycosides, saponins, steroids, terpenoids, tannins, alkaloids and phenolic compounds in the plant extract [71]. In 2005, three phenylethanoid glycosides, desrhamnosyl acteoside (**100**), acteoside (**101**) and poliumoside (**102**), were isolated from the callus cultures of *B. cristata* [72], followed by two phenolic compounds, *p*-coumaric acid (**103**) and α-tocopherol (**104**), two flavonoidal compounds, luteoline (**105**) and 7-methoxyluteoline (**106**), and iridodial glycosides, barlerin (**107**) and schanshiside methyl ester (**108**) from the leaves of *B. cristata* (Figure 9) [73]. Besides these compounds, researchers have found 4-hydroxy-*trans*-cinnamate derivatives and oleanolic acid (**109**) in this plant [74]. Biological investigations of this plant showed anti-inflammatory [69,75], and antioxidant properties [76,77]. There are few phytochemistry and biological studies of this species; therefore, it seems to be interesting for further investigation of new lead products.

### 2.9. Justicia gendarussa Burm F.

*J. gendarussa* (Family: Acanthaceae) is a shade-loving, quick-growing, evergreen plant mostly found in moist areas. In Myanmar, it is commonly known as pha-wa-net and distributed in Bago, Kachin, Mandalay, Shan, Taninthayi and Yangon. *J. gendarussa* is native to China and is distributed widely across India, in tropical and subtropical areas of Asia, Sri Lanka and Malaysia [78]. In traditional medicinal systems, different parts of *J. gendarussa* have been used in a variety of diseases. The leaves and tender shoots are diaphoretic and used for chronic rheumatism. The juice of the fresh leaves is dropped into the ear for treating earache and an oil prepared from the leaves is used topically to treat oedema caused by beriberi. The plant has been used by the native medical practitioners and tribes to treat various ailments including liver disorders, tumours, inflammation and skin diseases [79].

Two diphyllin glycosides were obtained from the stems and the barks of *J. gendarussa*; justiprocumins A–B (**110**,**111**) [80] (Figure 10). Souza et al. isolated four alkaloids, brazoides A–D (**112**–**115**), squalene, β-sitosterol and lupeol from the ethanol extract of leaves of *J. gendarussa* [81].

Patentiflorin A (**116**) from *J. gendarussa* displayed anti-HIV activity against a broad spectrum of HIV strains with IC_50_ values in the range of 24–37 nM, higher than the clinically used first anti-HIV drug, zidovudine AZT (IC_50_ 77–95 nM) [82]. In addition, justiprocumin B (**111**) showed potent activity against a broad spectrum of HIV strains with IC_50_ values in the range of 15–21 nM (AZT, IC_50_ 77–95 nM), nevirapine-resistant isolate HIV-1N119 with an IC_50_ value of 495 nM and AZT-resistant isolate HIV-11617-1 with (IC_50_ 185 nM). The compound also showed potent inhibitory activity against the NRTI-resistant isolate HIV-1 of the analogue AZT as well as the NNRTI-resistant isolate (HIV-1N) of the analogue nevaripine [80]. *J. gendarussa* had been reported to have a variety of pharmacological actions like hepatoprotective [83], antioxidant [84,85], anti-inflammatory and analgesic [85], antiangiogenic [86], anti-anxiety [87], anti-arthritic [88], anti-cancer [89,90], antibacterial [91], antifungal [92] and anthelmintic [93] effects based on folk medicine uses. The ethyl acetate soluble fraction isolated from the methanolic extract of the roots of *J. gendarussa* possessed anti-inflammatory activity in the carrageenan-induced paw edema model [94]. Scientific evidence for these pharmacological actions has not yet been found. It seems that *J. gendarussa* has the potential for being selected for further clinical studies. Moreover, relatively few chemical constituents of this plant are studied, hence it might be of interest to search for new bioactive compounds in its extracts.

### 2.10. Premna integrifolia L.

The plant *P. integrifoia* belongs to the family Verbenaceae, widely distributed in Vietnam, Laos and Cambodia [95]. In Myanmar, it is locally known by the names taung-tangyi or kywe-thwe and widely distributed in Mandalay, Rakhine and Taninthayi [4]. Traditionally, roots and stem barks are used as a laxative, carminative, and stomachic. Decoctions of the whole plant are used for fevers, rheumatism and neuralgia. The roots are used in the treatment of diabetes, inflammation, swelling, bronchitis, dyspepsia, liver disorders, piles, constipation and fever [96]. The decoction of the roots is used in gonorrhea and during convalescence from fever [97]. Two alkaloids, premnine and ganiarine, are isolated from the stem bark of *P. integrifolia* [97]. It is widely used by the traditional practitioners as a cardiotonic, antibiotic, anti-coagulant, stomachic, carminative, hepatoprotective and antitumor agent [98].

Two iridoid glycosides along with other ten compounds were isolated by Hang et al., in 2008 from the flowers: premnacorymboside A (**117**) (Figure 11), 10-*O*-*trans*-*p*-methoxycinnamoylcatalpol (**119**), and verbascoside, from the leaves: premnacorymboside B (**118**), 10-*O*-*trans*-*p*-methoxy-cinnamoylcatapol, scutellarioside II (**120**), premnaodoroside A (**121**), 1-*O*-*trans*-*p*-coumaroyl-α-l-rhamnopyranoside, hexyl glucoside, 4-hydroxy-2-methoxybenzaldehyde, and 4-hydroxy-benzaldehyde, from the stem bark of *P. integrifolia*: premnaodoroside A, scutellarioside II, quercetin-3-rutinoside, and leonuriside A [99]. Three diterpenoids, 1β,3α,8β-trihydroxy-pimara-15-ene (**122**), 6α,11,12,16-tetrahydroxy-7-oxoabieta-8,11,13-triene (**123**) and 2α,19-dihydroxypimara-7,15-diene (**124**) were identified from the root bark of *P. integrifolia* [100]. Major iridoid glycosides, 10-*O*-*trans*-*p*-methoxycinnamoylcatapol, 4″-hydroxy-*E*-globularinin (**125**) and premnosidic acid (**126**) [101] and two furofuran lignans, premnadimer (**127**) and 4β-hydroxyasarinin-1-*O*-β-glucopyranoside (**128**) together with other nine compounds were isolated from the stem bark of *P. integrifolia* and structurally elucidated by Yadav et al. [102].

Premnadimer (**127**), 4β-hydroxyasarinin-1-*O*-β-glucopyranoside (**128**), and iridoids 10-*O*-*trans*-*p*-methoxycinnamoylcatapol (**119**, IC_50_ 0.37 μM/mL), 4″-hydroxy-*E*-globularinin (**125**, IC_50_ 0.37 μM/mL), premnosidic acid (**126**) and 10-*O*-*trans*-*p*-coumaroyl-6-*O*-α-l-rhamnopyranosylcatapol showed antioxidant activity in the DPPH and NO free radical scavenging assays. Compound **126** and 10-*O*-*trans*-*p*-coumaroyl-6-*O*-α-l-rhamnopyranosylcatapol displayed a maximum ferric reducing ability in the FRAP assay [101]. Twenty-nine compounds representing 94.81% of the leaf oil were analysed by GC-MS. Among these, phytol (27.25%), α-humulene (14.21%), spathulenol (12.12%), 1-octen-3-ol (8.21%), eugenol (6.69%), phenylethyl alcohol (5.81%) and caryophyllene oxide (2.6%) were the major compounds. The major components of the oil from *P. integrifolia* and extracts of the leaves of this plant displayed great antibacterial activity potential against some bacterial strains such as *Sarcina lutea*, *Bacillus subtilis*, *Escherichia coli*, *Pseudomonas* sp., *Klebsiella pneumoniae* and *Xanthomonas campestris* [103]. In addition, the methanolic extract of *P. integrifolia* bark was reported to have antidiabetic activity along with antioxidant activity in vitro [104] and anti-inflammatory activity [105]. The traditional uses of the plant still need to be investigated scientifically.

### 2.11. Vitex trifolia L.

*V. trifolia* belongs to the Verbenaceae family, a small shrub, is distributed widely in Southeast Asia, Micronesia, Australia, East Africa [106], and Yunan, Guangxi, and Guangdong provinces in mainland China [107]. It is commonly known in Myanmar as kyaung-ban. In Myanmar, dried fruit powder is orally taken with honey in a dose of 5–10 gm for diarrhoeal diseases. For menstrual disorders and urinary disorders, indigestion, the dried powder is orally taken with roasted common salt and warmed water in the same dose as above. Leaves is used as a curry ingredient. Root powder is slurried with alcohol and externally used for muscle cramps.

From the fruits of *V. trifolia*, Ono reported isolation of an abietane-type diterpene, vitetrifolin A (**129**) (Figure 12), two labdane-type diterpenes, vitetrifolins B (**130**) and C (**131**), and diterpenes, rotundifuran (**132**), dihydrosolidagenone (**133**) and abietatriene 3β-ol (**134**) [108]. Moreover, Ono’s group also reported halimane-type diterpenes, vitetrifolins D–G (**135**–**138**) from the fruits of this plant [109]. Fruits of *V. trifolia* were found to contain five labdane-type diterpenes, vitexilactione, (rel 5*S*,6*R*,8*R*,9*R*,10*S*)-6-acetoxy-9-hydroxy-13(14)-landen-16,15-olide, rotundifuran, vitetrifolin and vitetrifolin E [110] and flavonoids, persicogenin (**139**), artemetin (**140**), luteolin (**141**), penduletin (**142**), vitexicarpin (**143**) and chrysosplenol-D (**144**) [111]. A series of labdane-type diterpenoids, vitextrifolins A–G (**145**–**151**), together with rotundifuran, vitetrifolin H, vitexilactone, vitexilactone B, deacetylvitexilactone, viteagnusin I, previtexilactone and negundol were isolated from the fruits of *V. trifolia* [112]. Additionally, an iridoid agnuside (**152**) was found in the leaves of the same species [113].

Negundoside (**153**), agnuside (**152**) and 6-*p*-hydroxybenzoyl mussaenosidic acid (**154**) isolated from the aerial parts of *V. trifolia* showed significant antioxidant activity with IC_50_ values of 9.96, 9.81 and 10.31 μg, respectively in DPPH and 16.25, 12.90 and 13.51 μg in NO radical scavenging assays [114]. The ethanol extract of the leaves of *V. trifolia* displayed free radical scavenging activity [113]. The methyl-*p*-hydroxybenzoate (**155**), which is reported from *V. trifolia* possessed potential mosquito larvicidal activity with the LC_50_ values of 5.77 ppm against *Culex quinquefasciatus* and 4.74 ppm against *Aedes aegypti* [115].

Essential oils of fresh leaves of *V. trifolia*, which was extracted by hydro-distillation in a Clevenger’s apparatus, were evaluated against Vth instar larvae of *Spilosoma obliqua*, for insect growth regulatory activity [116]. According to Matsui et al. the leaves of *V. trifolia* showed the significant inhibitory activity on interleukin (IL)-1-β, IL-6 and inducible nitric oxide synthase (iNOS) mRNA synthesis and have a little effect on tumor necrosis factor (TNF) [117] and it acts on inflammatory activity related gene expression by inhibiting the NF-κB action [118]. The labdane-type diterpenoids [110] and flavonoids from the fruits of this plant [111] displayed anti-proliferative effects on cancer cells through the induction of the apoptosis and the inhibition of the cell cycle. Biological assays have been proved that the hexane and dichloromethane extracts of this plant are highly cytotoxic against several cancer cell lines [119]. The biological activities of *V. trifolia* have been reported scientifically, however, some traditional uses are still uninvestigated.

### 2.12. Acacia pennata (L.) Willd.

*A. pennata*, belonging to the family Mimosaceae, is a large woody prickly climber with bipinnate leaves. *A. pennata* (local name in Myanmar suyit or suboke-gyi) is widely distributed in regions of south and southeast Asia including Bangladesh, Butan, India, Sri Lanka, southwest China and Thailand [120]. In Myanmar, barks of *A. pennata* have been used in the treatment of asthma and bronchitis. It is reported to have some medicinal value. The leaf juice when mixed with milk is given to infants for indigestion. The *A. pennata* plant is used as an antiseptic for burning urine and for curing bleeding gums. The root bark can be used as antiflatulent and to cure stomach pain. It is also used in the treatment of bronchitis, cholera and asthma. A decoction of the leaves is used for general treatment of body aches, headaches and fevers [121].

From the leaves of *A. pennata*, two diterpenoids, a flavonoid glycoside together with another compounds were discovered: taepeenin D (**156**) (Figure 13), (+)-drim-8-ene (**157**), quercetin 3-*O*-β-d-glucopyranosyl-4-*O*-β-d-glucopyranoside (**158**), labdanolic acid and 8,15-labdanediol [122], quercetin 4′-*O*-α-l-rhamnopyranosyl-3-*O*-β-d-allopyranoside (**159**), apigenin 6-*C*-[2″-*O*-(E)-feruloyl-β-d-glucopyranosyl]-8-*C*-β-glucopyranoside (**160**), together with isorhamnetin 3-*O*-α-l-rhamno-pyranoside (**161**), kaempferol 3-*O*-α-l-rhamnopyranosyl-(1→4)-β-d-glucopyranoside (**162**), and isovitexin (**163**) [123]. From the aerial parts of *A. pennata*, a species distributed in Popa Mountain National Park (Mandalay, Myanmar), five flavonoid glycosides; (2*R*,3*S*)-3,5,7-trihydroxyflavan-3-*O*-α-l-rhamnopyranoside (**164**), (2*S*)-5,7-dihydroxyflavan-7-*O*-β-d-glucopyranoside-(4α→8)-epi-afzelechin-3-*O*-gallate (**165**), (2*R*)-4′,7-dihydroxyflavan-(4α→8)-(2*R*,3*S*)-3,5,7-trihydroxyflavan-3-*O*-α-l-rhamnopyranoside (**166**), 5,7-dihydroxyflavone-6-*C*-β-boivinopyranosyl-7-*O*-β-d-gluco-pyranoside (**167**), and 5,7-dihydroxyflavone 7-*O*-β-d-glucopyranosyl-8-*C*-β-boivinopyranoside (**168**), along with other compounds, quercetin-3-*O*-β-d-glucopyranoside (**169**), quercetin-3-*O*-α-l-rhamnopyranoside (**170**), chrysin-7-*O*-β-d-glucopyraniside (**171**), kaempferol 3-*O*-α-l-rhamnopyranoside (**172**), koaburanin (**173**) and pinocembrin-7-*O*-β-d-glucopyranoside (**174**) were isolated [124].

Taepeenin D (**156**), (+)-drim-8-ene (**157**) and quercetin 3-*O*-β-d-glucopyranosyl-4-*O*-β-d-glucopyranoside (**158**) showed cytotoxic activity against human pancreatic (PANC1) cancer with IC_50_ values of 3.2 μM, 15.1 μM and 26.6 μM, respectively, and prostate cancer (DU145) cells, with IC_50_ values of 3.4 μM, 23.2 μM and 30.0 μM, respectively, and no toxic effects on normal cells. Taepeenin D (**156**), (+)-drim-8-ene (**157**) and quercetin 3-*O*-β-d-glucopyranosyl-4-*O*-β-d-glucopyranoside (**158**), which possessed Hedgehog/GLi-mediated transcriptional activity, showing IC_50_ values of 1.6, 13.5 and 10.5 μM [122]. Biological activities of *A. pennata* were reported to include anti-parasitic [117] anti-nociceptive and anti-inflammatory [123,125], and antioxidant effects [126]. The phytochemical constituents and biological activities of *A. pennata* have been reported, however, further pharmacological studies are still needed to scientifically prove the basis of their actions.

### 2.13. Cassia auriculata Linn.

*C. auriculata* belongs to the family Caesalpiniaceae, which is an ethno-botanically important shrub with attractive yellow flowers. It is cultivated in Myanmar and commonly known as peik-thin-khet. The ethanolic extract of the roots of *C. auriculata* displayed the nephroprotective activity in cisplatin and gentamicin induced renal injury in male albino rats [127]. The plant has been reported to possess antipyretic, hepatoprotective, antidiabetic, antiperoxidative and antihyperglyceamic and microbicidal activity [128,129]. The flowers are used to treat urinary discharges, nocturnal emissions, diabetes and throat irritation. The leaves are anthelmintic and good for ulcers, skin diseases and leprosy [130].

Nakamura reported a benzocoumarin glycoside, avaraside I (**175**, Figure 14), along with other compounds such as av170araol I (**176**), luteolin, kaempferol, quercetin (**177**), myricetin (**178**), 3-methoxy-luteolin (**179**), kaempferol 3-*O*-β-d-glucopyranoside (**180**), isoquercetin (**181**), myricetin-3-*O*-β-d-glucopyranoside (**182**), kaempferol-3-*O*-rutinoside (**183**), rutin (**184**), myricetin 3-*O*-rutinoside (**185**), lanceolatin B (**186**), pseudosemiglabrin (**187**), (+)-catechin, (+)-gallocatechin, (−)-epicatechin, 6-demethoxycapillarisin, 6-demethoxy-7-methylcapillarisin, (2*S*)-7,4-dihydroxyflavan(4β→8)-catechin (**188**), (2*S*)-7,4-dihydroxyflavan(4β→8)-gallocatechin (**189**), (2*S*)-7,4-dihydroxyflavan(4β→8)-epicatechin (**190**), (2*S*)-7,4-dihydroxyflavan(4β→8)-epigallocatechin (**191**), chrysophanol, emodin, physcion, roseoside, bridelionoside C, benzyl-*O*-β-d-apiofuranosyl (1→2)-β-d-glucopyranoside and (−)-epigallocatechin (**192**) from the leaves of *C. auriculata* [131].

The ethanolic extract of *C. auriculata* flowers showed antihyperlipidemic activity [132]. The aqueous extract of the leaves of *C. auriculata* possessed antihyperglycemic and hypolipidemic activity in streptozotocin (STZ)-induced mild diabetic and severe diabetic rats [133]. The methanol extract of the root of this plant had significant hepatoprotective activities and antitubercular drug-induced hepatotoxicity [134]. Pseudosemiglabrin (**187**) [inhibition %: 37.0 ± 3.7 (*p* < 0.01) at 30 μM], (2*S*)-7,4-dihydroxyflavan (4β→8)-catechin (**188**) [inhibition %: 33.5 ± 2.9 (*p* < 0.01) at 30 μM], (2*S*)-7,4-dihydroxyflavan (4β→8)-gallocatechin (**189**) [inhibition %: 28.2 ± 4.7 (*p* < 0.01) at 30 μM], and (−)-epigallocatechin (**192**) [inhibition %: 24.4 ± 3.1 (*p* < 0.01) at 30 μM], isolated from *C. auriculata* displayed hepatoprotective effects against cytotoxicity induced by D-GalM in primary cultured mouse hepatocytes [135]. *C. auriculata* leaf extract has a protective action against alcohol-induced oxidative stress to the cells as evidenced by the lowered tissue lipid peroxidation and elevated levels of the enzymic and non-enzymic antioxidants [135]. Polyphenols derived from the flowers of *C. auriculata* showed immunomodulatory effects on various immune cell compositions and their relevant functions [136]. The biological activities of chemical constituents and active principle(s) from this plant are still lacking thorough investigations.

### 2.14. Croton oblongifolius Roxb.

*C. oblongifolius* (Euphorbiaceae), is popularly known as thetyin-gyi in Mynanmar. It is a medium sized tree widely distributed in the region of Mount Popa in Myanmar. Seeds are used as diarrhea and oedema, and very useful for inflammation, either taken orally or as an external application in Myanmar. Root bark and seeds are reported to be used as purgative, liver diseases and high blood pressure. It is used in gastric ulcers; the bark is used to treat dyspepsia and the roots to treat dysentery [137].

From the leaves of *C. oblongifolius*, a clerodane type diterpene, nasimalun A (**193**) (Figure 15) was isolated [138]. Nasimalun A (**193**) from the leaves of *C. oblongifolius* displayed moderate cytotoxicity toward MOLT-3 cell line with IC_50_ value of 12.0 μM. In addition, this compound showed antibacterial activity showing the MIC values of 50, 12.5, and 100 μg/mL for *Bacilus cereus* and both *Staphylococcus aureus*, and *Staphylococcus epidermidis* respectively [139]. Clerodane type diterpenes, crovatin (**194**), methyl 15,16-epoxy-3,13(16),14-ent-clerodatrien-18,19-olide-17-carboxylate (**195**) and dimethyl 15,16-epoxy-12-oxo-3,13(16),14-ent-clerodatriene-17,18-dicarboxylate (**196**), nasimaluns A and B (**197**), levatin, (−)-hardwickiic acid, 15-hydroxy-*cis-*ent-cleroda-3,13(*E*)-diene (**198**) and a sesquiterpene, patchoulenone were reported from the roots of *C. oblongifolius* [140]. Two cleidtanthane-type diterpenoids, hydroxycleistantha-13(17),15-ene (**199**) and 3,4-seco-cleistantha-4(18),13(17),15-triene-3-oic acid (**200**) [141], four furanocembranoids 1–4 (**201**–**204**) [142], (−)-ent-kuar-16-en-19-oic acid (**205**) [138], furoclerodane-type diterpene, croblongifolin (**206**), one clerodane, crovatin, eight labdane-types, nidorellol (**207**), 2-acetoxy-3-hydroxylabda-8(17),12(*E*)-14-triene (**208**), 3-acetoxy-2-hydroxylabda-8(17),12(*E*),14-triene (**209**), 2,3-dihydroxylabda-8(17),12(*E*),14-triene (**210**) [143], labda-7,12(*E*),14-triene (**211**), labda-7,12(*E*),14-triene-17-al (**212**), labda-7,12(*E*),14-triene-17-ol (**213**) and labda-7,12(*E*),14-triene-17-oic acid (**214**) [144], three cembranoids, neocrotocembranal (**215**) [145], crotocembraneic acid (**216**) and neocrotocembraneic acid (**217**) [146], one diterpene acid, (+)-isopimara-7(8),15-diene-19-oic acid [147], oblongifoliol (**218**) [148] were obtained from the stem bark of *C. oblongifolius*.

Croblongifolin (**206**) isolated from the stem bark of *C. oblongifolius* was displayed cytotoxicity towards several human tumor cell lines including HepG2 (IC_50_ 0.35 μM), SW620 (IC_50_ 0.47 μM), CHAGO (IC_50_ 0.24 μM), KATO3 (IC_50_ 0.35 μM) and BT474 (IC_50_ 0.12 μM) [143]. Neocrotocembranal (**215**) was also reported cytotoxicity against P-388 cells in vitro with an IC_50_ 6.48 μg/mL and inhibited platelet aggregation induced by thrombin [145]. (−)-ent-kuar-16-en-19-oic acid (**205**) was possessed the inhibition of Na^+^, K^+^-ATPase activity with an IC_50_ of 2.2 × 10^−5^ M [138]. Furanocembranoid 1 (**201**) proved to be cytotoxic for human tumor cell lines BT474, CHAGO, Hep-G2, KATO-3 and SW620s, with IC_50_ values of 7.8, 7.0, 5.6, 5.9 and 6.3 μg/mL, respectively by MTT assay. Compound (**203**) showed inhibitory activity against BT474 cell lines (IC_50_ 9.6 μg/mL), CHAGO (IC_50_ 7.1 μg/mL), Hep-G2 (IC_50_ 5.7 μg/mL), KATO-3 (IC_50_ 8.2 μg/mL) and SW-620s (IC_50_ 5.6 μg/mL), while (**204**) inhibited the growth of the human tumor BT474 (IC_50_ 9.6 μg/mL), CHAGO (IC_50_ 9.3 μg/mL), Hep-G2 (IC_50_ 6.1 μg/mL), KATO-3 (IC_50_ 8.1 μg/mL) and SW-620s (IC_50_ 6.0 μg/mL) cell lines [142]. In 2016, Adiga evaluated the anti-tumor activity of the methanolic extract of the stem bark of *C. oblongifolius* [148].

The methanolic extract of the aerial part of *C. oblongifolius* showed maximum antihepatotoxic activity, whereas, the other extracts (petroleum ether and acetone) showed lower activity [149]. Phytochemical constituents and biological activities of *C. oblongifolius* have been reported scientifically.

### 2.15. Glycosmis pentaphylla Correa

*G. pentaphylla* is a shrub or small tree, ca. 1.5–5.0 m high, and is widely distributed from India, Malaysia, and Southern China to the Philippine Islands [150]. It is commonly known as taw-shauk or obok in Myanmar and is widely distributed in Bago, Magway, Mandalay and Yangon. In traditional Myanmar medicine, juice of the leaves of *G. pentaphylla* is used in fevers, liver complaints and as a vermifuge. A paste of the leaves mixed with ginger is applied for eczema and skin affections. A decoction of the roots is given for facial inflammation. *G. pentaphylla* is a species of plants belonging to the Rutaceae family, which has been used in several countries as a traditional medicinal plant for the treatment of various ailments. It is used for the treatment of cough, rheumatism, anaemia, and jaundice. Stems and roots of this plant are used for treatment of ulcer [151].

A naphthoquinone, glycoquinone (**219**) and acridone alkaloid, glycocitrine-III (**220**) (Figure 16) [152], four hydroquinone diglycoside acyl esters, glypentosides A–C (**221**–**223**) and seguinoside F (**224**) [153], carbazole alkaloid, 4-(7-hydroxy-3-methoxy-6-methyl-9*H*-carbazol-4-yl)but-3-en-2-one (**225**), two dimeric carbazole alkaloids, bisglybomine B (**226**) and biscarbalexine A (**227**), together with glybomine B, glycoborinine, carbalexine A, 4,8-dimethoxy-1-methyl-3-(3-methylbut-2-en-1-yl)-quinolin-2(1*H*)-one, 4,8-dimethoxyfuro [2,3-*b*]quinoline, skimmianine and arborinine [154], three phenolic glycosides, glycopentosides D–F (**228**–**230**) [155] and two carbazole indole-type dimeric alkaloids, glycosmisines A (**231**) and B (**232**) [156] were isolated from the stems of *G. pentalphylla*. Alkaloid, glycophymine (**233**) and an amide, glycomide from the flowers of this species [157] and derivative of carbazole alkaloid, glycozolinine (**234**) was identified from the seeds of *G. pentaphylla* [158]. In 1966, alkaloid, skimmianine (**235**) and acridone alkaloids, noracronycine (**236**), des-*N*-methylacronycine (**237**) and des-*N*-methylnoracronycine (**238**) were isolated from the root bark of *G. pentaphylla* [159] and followed by carbazole (**239**) and 3-methylcarbazole (**240**) [160] from the same parts of this species in 1987. Bhattacharya obtained the carbazole alkaloids, glycozolidol (**241**) [161], glycozolidal (**242**) [162] from the roots of *G. pentalphylla*. One quinolone alkaloid, glycolone [163], three furanopyridine alkaloids, glypenfurans A-C, along with skimmianine, γ-fagarine, *R*-(+)-platydesmine, melicarpine, robustine and haplopine [164] were isolated from the leaves of the same plant. The hydroperoxyquinolone alkaloid, glycopentaphyllone, acutifolin, 3-(3′,3′-dimethylallyl)-4,8-dimethoxy-*N*-methylquinolin-2-one, glycocitlone C, arborine, dictamine, γ-skimmianine, 1-hydroxy-3,4-dimethoxy-10 methylacridan-9-one and arborinine were reported from the fruits of *G. pentaphylla* [165].

The methanolic extract of the leaves and stems of *G. pentaphylla* showed antimicrobial, antioxidant and cytotoxic effects [166]. The acridone alkaloid arborinine, isolated from *G. pentaphylla*, showed significant inhibitory activity towards the growth of crown gall tumors produced by *Agrobacterium tumefaciens* in a potato disc bioassay [167]. The methanolic extract of the leaves of *G. pentaphylla* showed hepatoprotective activity against paracetamol-induced hepatotoxicity in Swiss albino mice [168]. Streejith et al., reported in 2012 the anti-hepatocellular carcinoma cell line activity of *G. pentaphylla* ethanol extract in reducing the proliferation of Hep3 B cells [151]. Biscarbalexine A (**227**), glycosmisines A (**231**) and B (**232**) displayed significant the cytotoxic activity against human cancer A549, HepG-2 and Hun-7 cell lines, showing with an IC_50_ values of 56.05, 43.68 and 57.01 μM for A549 cell line, 60.06, 50.30 and 62.89 μM for HepG-2 cell line and 73.16, 30.60 and 62.87 μM for Huh-7 cell line, respectively [156]. *G. pentaphylla* ethanolic crude extract showed a significant antioxidant activity [169]. According to the literature, this plant seems to be a rich source of bioactive compounds. Various biological and pharmacological activities of selected medicinal plants and their products have been summarized in Table 2.

## 3. Conclusions

In this review, we have systematically summarized the structures and properties of 242 compounds isolated from 15 species of selected Myanmar medicinal plants. Among the 15 selected medicinal plants that have been described in this review, *Erythrina suberosa* might have potential as a future anticancer therapeutic based on the literature. The plants, *Croton oblongifolius*, *Eriosema chinense*, *Premna integrifolia* and *Erythrina suberosa*, their phytochemical constituents and biological activities are extensively investigated. Studies on the plants of *Acacia pennata*, *Cassia auriculata*, *Glycomis pentaphylla* and *Tadehagi triquetrum* have reported the presence of several bioactive compounds whose biological and pharmacological applications can further be investigated. The chemical and pharmacological action of *Vitex trifolia* have mainly emphasized on fruits, however, the chemical constituents of other parts of this plant and some traditional uses are still missing. On the other hand, the biological activities of the various extracts of *Sesbania grandiflora*, and *Andrographis echioides* have been reported scientifically but there have been a few reports on chemical constituents. There are few phytochemistry and biological studies of *Dalbergia cultrata* and *Millettia pendula*, *Barleria cristata* and *Justicia gendarussa*; *Acacia pennata* and *Cassia auriculata*, therefore, they seem to be interesting for investigation of new pharmaceuticals.

A vast reservoir of selected species remains untapped in terms of phytochemical constituents, as well as pharmacology, and this is the research gap for future studies. Further investigations of endemic plants in Myanmar and their phytochemical constituents are necessary to completely understand the molecular mechanisms of their action in vivo and in vitro and to assure the plant extracts are safe for human use.

## Figures and Tables

**Figure 1 molecules-24-00293-f001:**
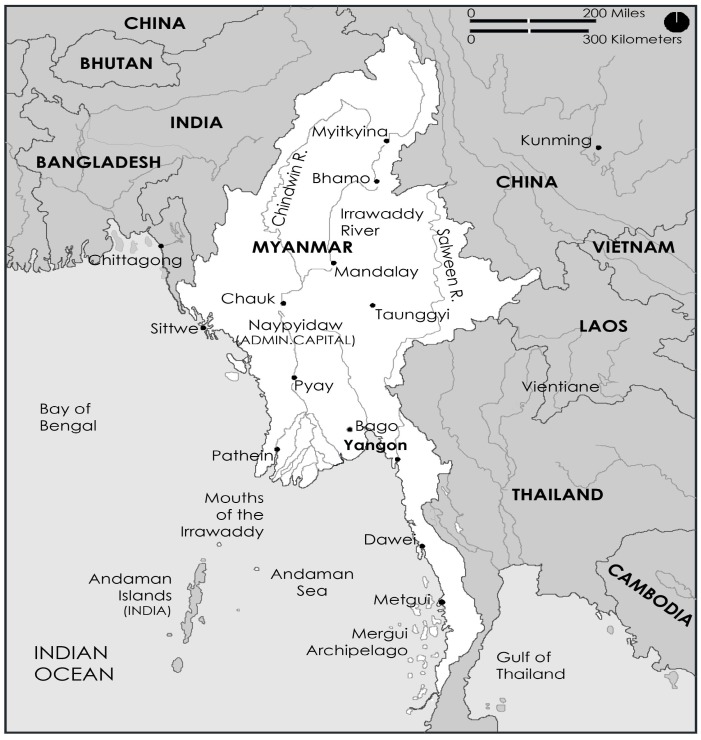
Map showing the Myanmar Region.

**Figure 2 molecules-24-00293-f002:**
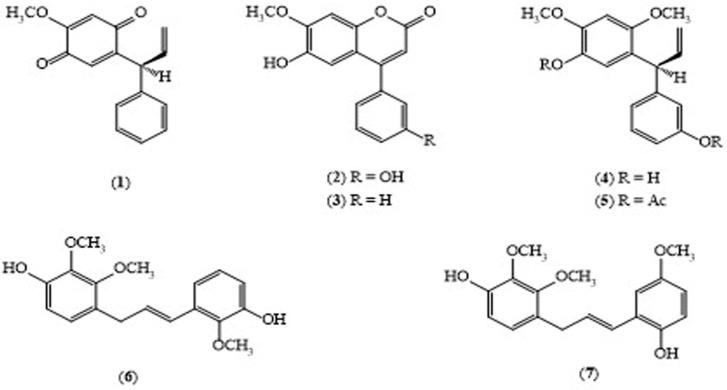
Structures of selected compounds isolated from *Dalbergia culatrata*.

**Figure 3 molecules-24-00293-f003:**
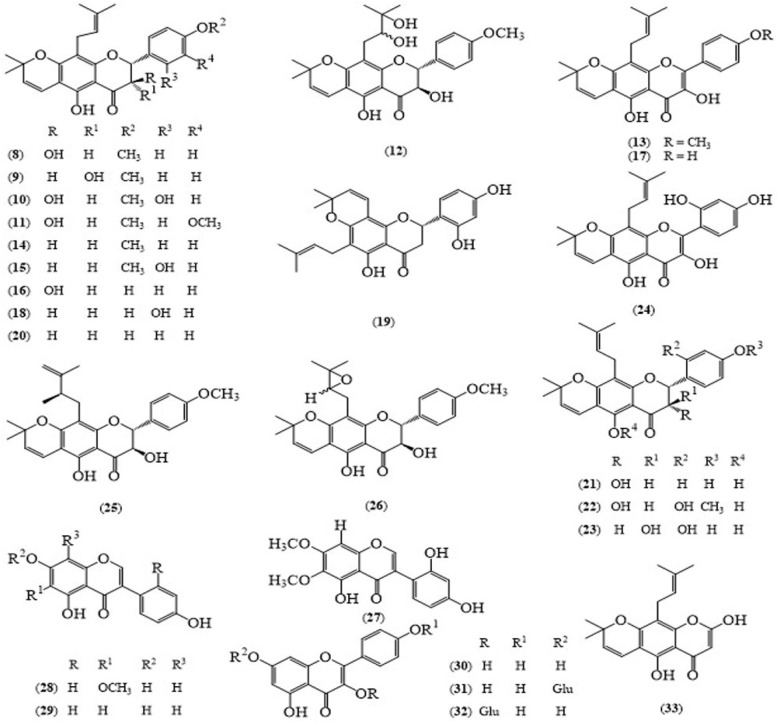
Structures of selected compounds isolated from *Eriosema chinense*.

**Figure 4 molecules-24-00293-f004:**
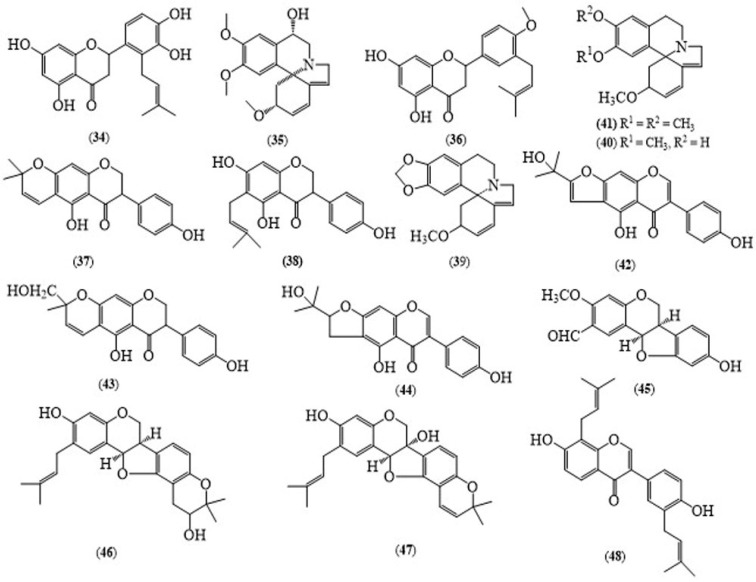
Structures of selected compounds isolated from *Erythrina suberosa*.

**Figure 5 molecules-24-00293-f005:**
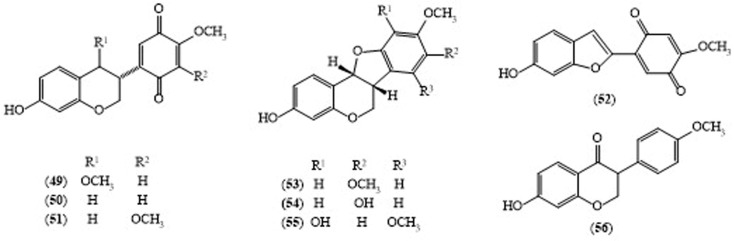
Structures of selected compounds isolated from *Millettia pendula*.

**Figure 6 molecules-24-00293-f006:**
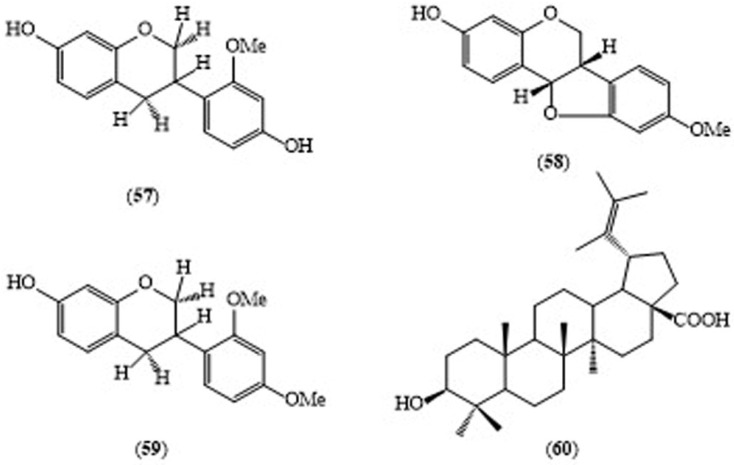
Structures of selected compounds isolated from *Sesbania grandiflora*.

**Figure 7 molecules-24-00293-f007:**
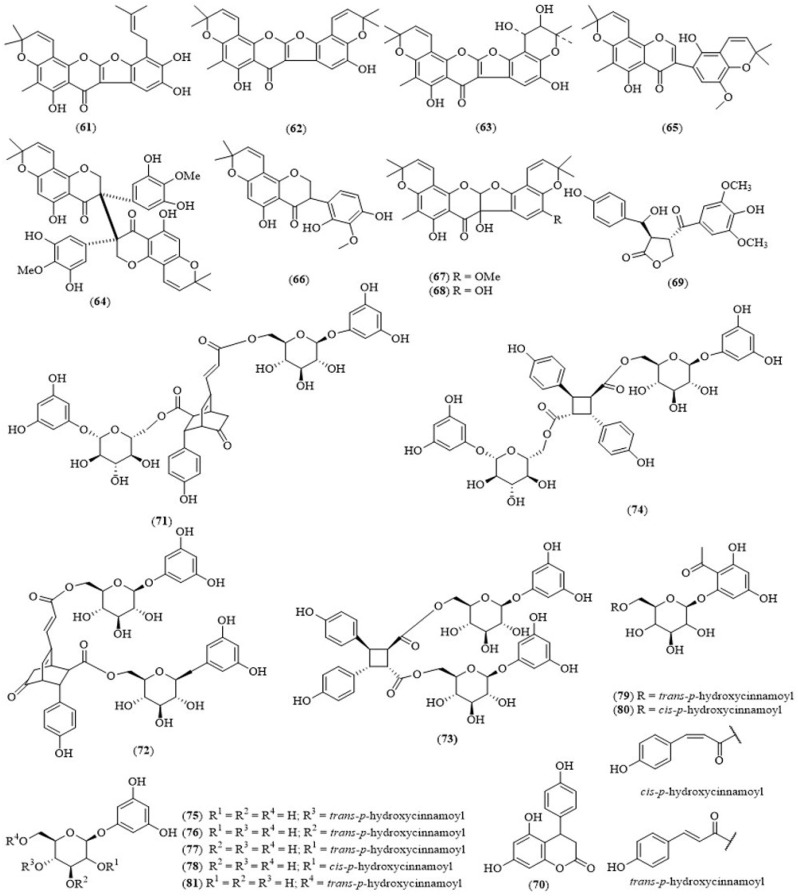
Structures of selected compounds isolated from *Tadehagi triquetrum*.

**Figure 8 molecules-24-00293-f008:**
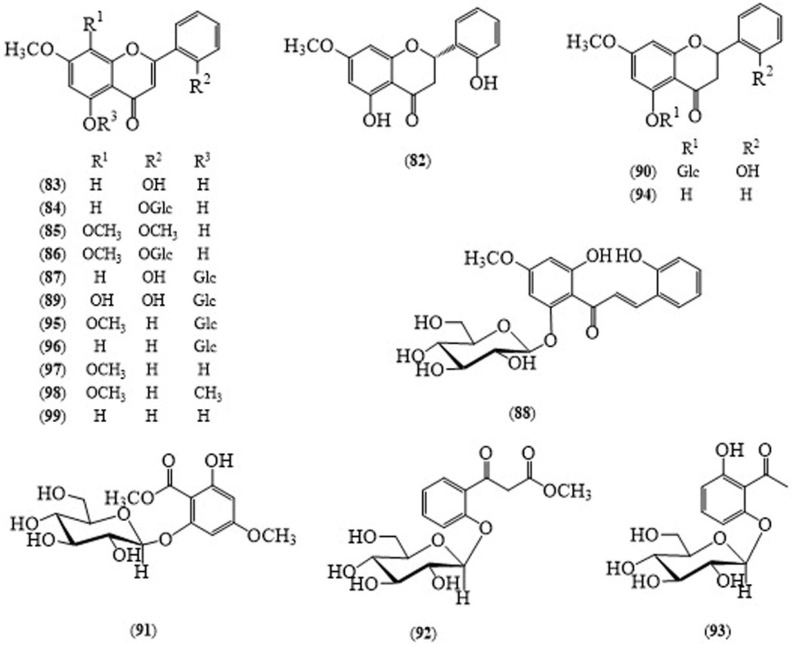
Structures of selected compounds isolated from *Andrographis echioides*.

**Figure 9 molecules-24-00293-f009:**
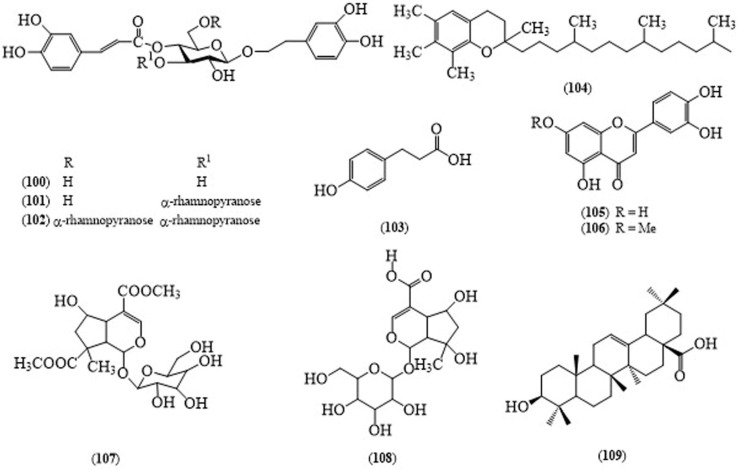
Structures of selected compounds isolated from *Barleria cristata*.

**Figure 10 molecules-24-00293-f010:**
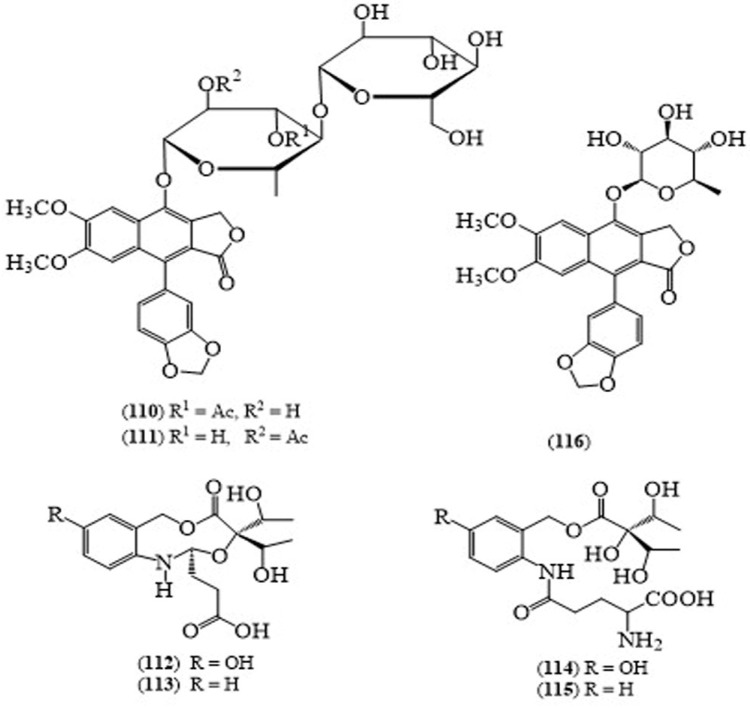
Structures of selected compounds isolated from *Justicia gendarussa*.

**Figure 11 molecules-24-00293-f011:**
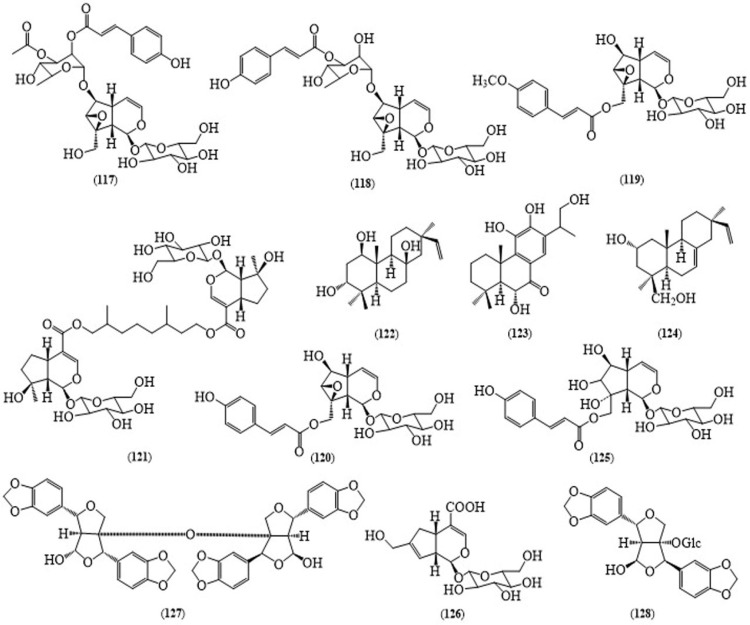
Structures of selected compounds isolated from *Premna integrifolia*.

**Figure 12 molecules-24-00293-f012:**
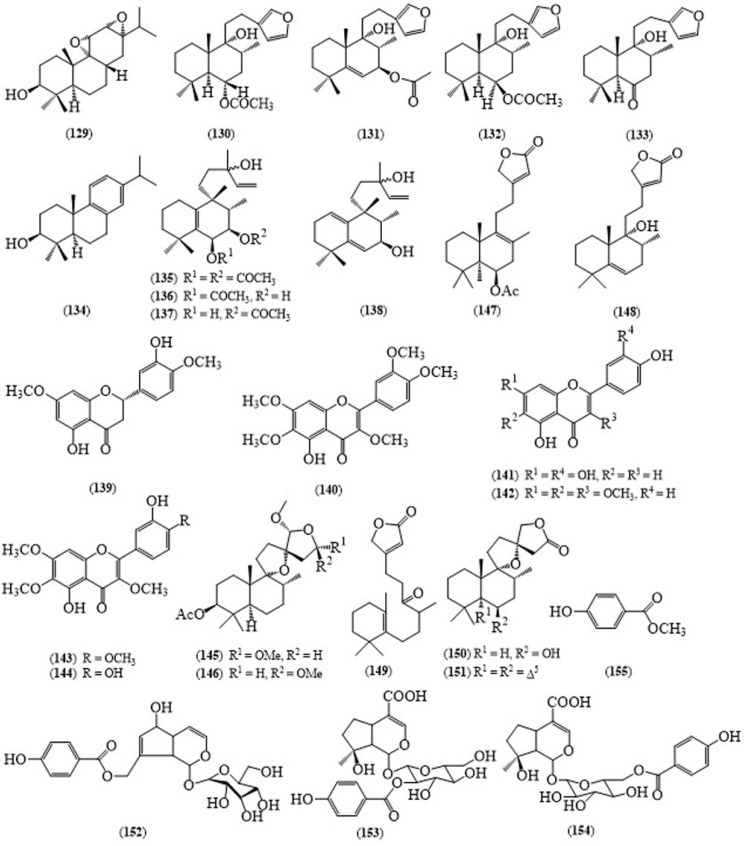
Structures of selected compounds isolated from *Vitex trifolia*.

**Figure 13 molecules-24-00293-f013:**
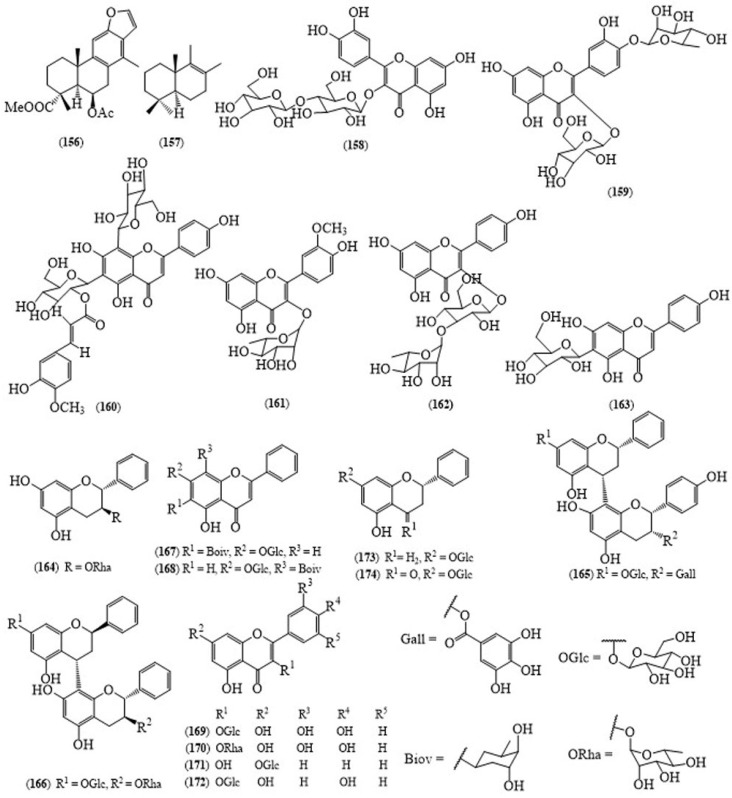
Structures of selected compounds isolated from *Acacia pennata*.

**Figure 14 molecules-24-00293-f014:**
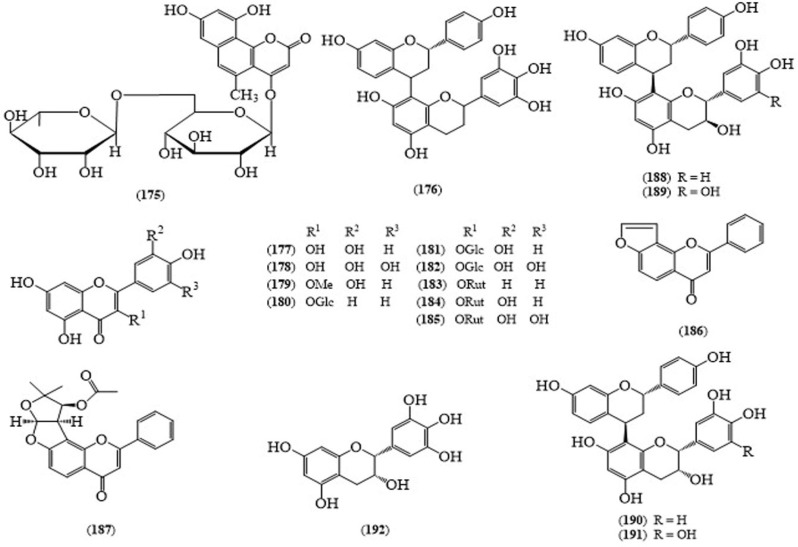
Structures of selected compounds isolated from *Cassia auriculata*.

**Figure 15 molecules-24-00293-f015:**
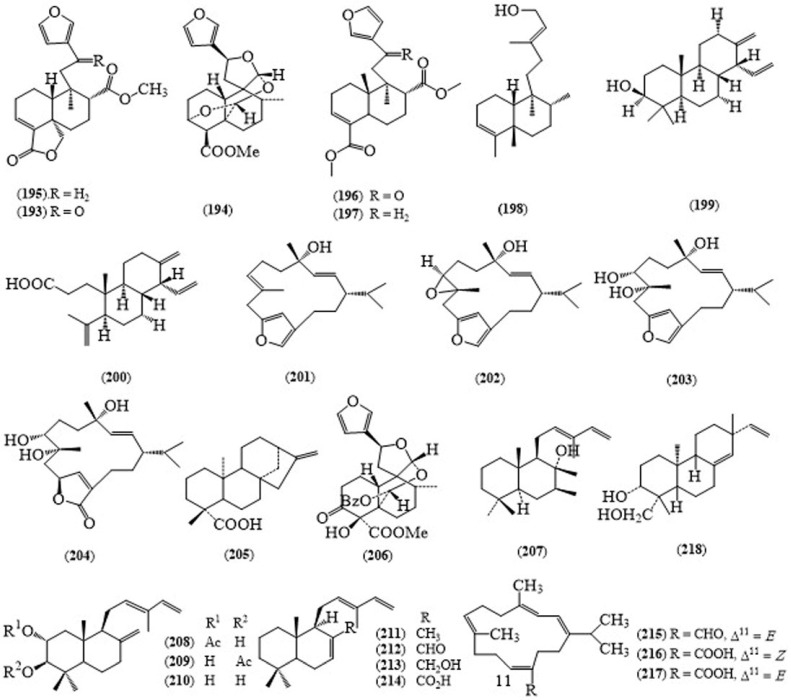
Structures of selected compounds isolated from *Croton oblongifolius*.

**Figure 16 molecules-24-00293-f016:**
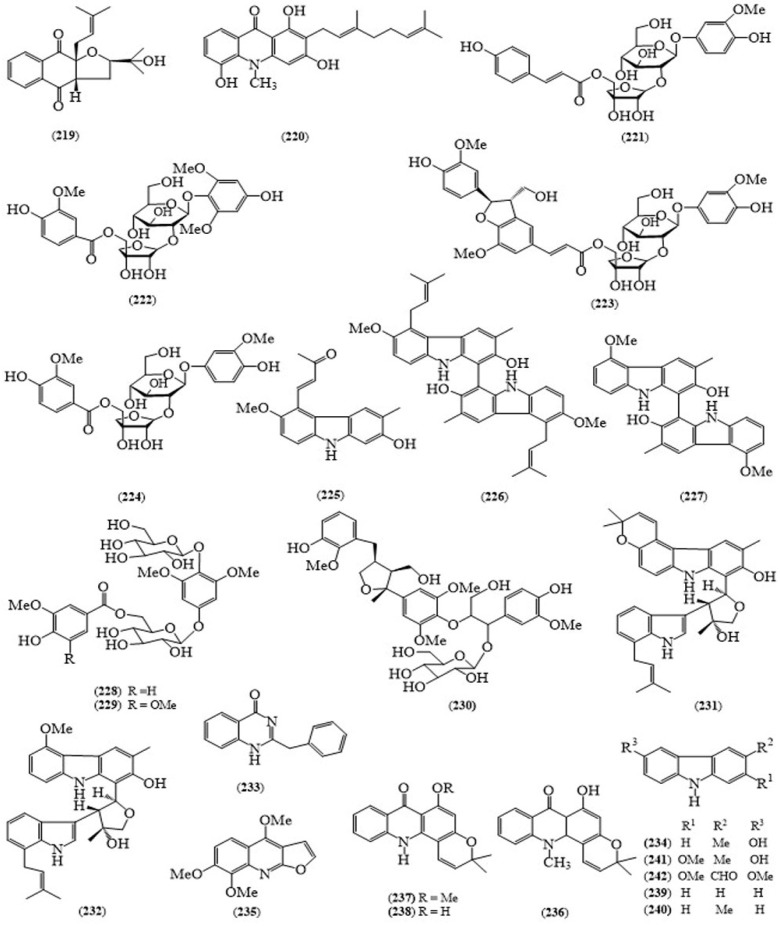
Structures of selected compounds isolated from *Glycosmis pentaphylla*.

**Table 1 molecules-24-00293-t001:** Scientific name, botanical family and local name of the selected Myanmar medicinal plants.

No	Scientific Name	Family Name	Local Name	Ref(s)
1	*Dalbergia cultrata*	Fabaceae	Yin-daik	[4]
2	*Eriosema chinense*	Fabaceae	Peik-san-gale	[4]
3	*Erythrina suberosa*	Fabaceae	Ka-thit	[4]
4	*Millettia pendula*	Fabaceae	Thin-win	[4]
5	*Sesbania grandiflora*	Fabaceae	Paukpan-phyu	[4]
6	*Tadehagi triquetrum*	Fabaceae	Lauk-thay or shwe-gu-than-hlet	[4]
7	*Andrographis echioides*	Acanthaceae	Sega-gyi-hmwe-tu	[4]
8	*Barleria cristata*	Acanthaceae	Leik-tha-ywe-pya	[4]
9	*Justicia gendarussa*	Acanthaceae	Pha-wa-net	[4]
10	*Premna integrifolia*	Verbenaceae	Taungtan-gyi	[4]
11	*Vitex trifolia*	Verbenaceae	Kyanung-ban	[4]
12	*Acacia pennata*	Mimosaceae	Suyit or Suboke-gyi	[4]
13	*Cassia auriculata*	Caesalpiniaceae	Peik-thingat	[4]
14	*Croton oblongifolius*	Euphorbiaceae	Thetyin-gyi	[4]
15	*Glycomis pentaphylla*	Rutaceae	Taw-shauk	[4]

**Table 2 molecules-24-00293-t002:** Biological/pharmacological activities of compounds isolated from selected medicinal plants.

Source	Compound	Biological/Pharmacological Activities	Reference(s)
*D. culrata* (Stem bark)	Dalberatin A (**6**)	Cancer chemopreventive activity with IC_50_ of 212 (mol ratio/32 pmol TPA)	[10]
*D. culrata* (Stem bark)	Dalberatin B (**7**)	Cancer chemopreventive activity with IC_50_ of 303 (mol ratio/32 pmol TPA)	[10]
*E. chinense* (Roots)	Khonklonginol A (**8**)	Cytotoxicity against KB (IC_50_ 3.1 μg/mL), NCI-H187 (IC_50_ 3.0 μg/mL), Antimycobacterial activity against *Mycobacterium tuberculosis* H37Ra (MIC 25 μg/mL), Antimicrobial activity against *Bacillus cereus* (MIC 150 μg/mL), *Staphylococcus agalactiae* (MIC 2.3 μg/mL) and *S. pyrogenes* (MIC 2.3 μg/mL), Antioxidant activity (IC_50_ 7.919 mM)	[11,12]
*E. chinense* (Roots)	Khonklonginol B (**9**)	Cytotoxicity against KB (IC_50_ 3.8 μg/mL), NCI-H187 (IC_50_ 4.3 μg/mL)	[11]
*E. chinense* (Roots)	Lupinifolinol (**16**)	Cytotoxicity against KB (IC_50_ 1.73 μg/mL), NCI-H187 (IC_50_ 3.5 μg/mL), Antimicrobial activity against *Candia albicans* (IC_50_ 75 μg/mL), *Bacillus cereus* (IC_50_ 4.7 μg/mL), *Listeria monocytogenes* (IC_50_ 9.4 μg/mL), *Staphylococcus aureus* (IC_50_ 75 μg/mL), *S. aureus*RASA (IC_50_ 9.4 μg/mL), *S. agalactiae* (IC_50_ 4.7 μg/mL), *S. epidermidis* (IC_50_ 9.4 μg/mL), *S. pyrogenes* (IC_50_ 2.3 μg/mL), Antioxidant activity (IC_50_ 1.768 mM)	[11,12]
*E. chinense* (Roots)	Dehydrolupinifolinol (**17**)	Antimycobacterial activity against *Mycobacterium tuberculosis* H37Ra (MIC 12.5 μg/mL)	[11]
*E. chinense* (Roots)	Flemichin D (**18**)	Antimycobacterial activity against *Mycobacterium tuberculosis* H37Ra (MIC 12.5 μg/mL), Antimicrobial activity against with the values of (IC_50_ ˃ 150 μg/mL) each for *Candida albicans*, *Escherichia coli*, *Klebsiella pneumoniae* and *Pseudomonas aeruginosa* and (IC_50_ 4.7 μg/mL) for each *Bacillus cereus*, *Enterococcus faecalis*, *Listeria monocytogenes*, *Staphylococcus aureus, S. aureus* RASA, *S. agalactiae*, *S. epidermidis* and *S. pyrogenes,* Antioxidant activity (IC_50_ 0.538 mM)	[11,12]
*E. chinense* (Roots)	Eriosemaone A (**19**)	Antimycobacterail activity against *Mycobacterium tuberculosis* H37Ra (MIC 12.5 μg/mL)	[11]
*E. chinense* (Roots)	Lupinifolin (**20**)	Antimycobacterial activity against *Mycobacterium tuberculosis* H37Ra (MIC 12.5 μg/mL)	[11]
*E. chinense* (Roots)	3-epi-lupinifolinol (**21**)	Antioxidant activity (IC_50_ 0.681 mM)	[12]
*E. chinense* (Roots)	2′-dihydroxy lupinifolinol (**23**)	Antimicrobial activity against *Candia albicans* (IC_50_ 37.5 μg/mL), *Escherichia coli* (IC_50_ 75 μg/mL), *Klebsiella pneumoniae* (IC_50_ 75 μg/mL), and *Pseudomonas aeruginosa* (IC_50_ 75 μg/mL), *Bacillus cereus* (IC_50_ 2.3 μg/mL), *Enterococcus faecalis* (IC_50_ 9.4 μg/mL), *Listeria monocytogenes* (IC_50_ 9.4 μg/mL), *Staphylococcus aureus* (IC_50_ 4.7 μg/mL), *S. aureus*RASA (IC_50_ 4.7 μg/mL), *S. agalactiae* (IC_50_ 4.7 μg/mL), *S. epidermidis* (IC_50_ 37.5 μg/mL), *S. pyrogenes* (IC_50_ 2.3 μg/mL), Antioxidant activity (IC_50_ 0.252 mM)	[12]
*E. chinense* (Roots)	3,5,2′,4′-Tetrahydroxy-6″,6″ dimethylpyrano (2″,3″:7,6)-8-(3‴,3‴-dimethylallyl) flavone (**24**)	Antimicrobial activity against *Candia albicans* (IC_50_ 75 μg/mL), *Escherichia coli* (IC_50_ ˃ 150 μg/mL), *Klebsiella pneumoniae* (IC_50_ 150 μg/mL), and *Pseudomonas aeruginosa* (IC_50_ 150 μg/mL), *Bacillus cereus* (IC_50_ 9.4 μg/mL), *Enterococcus faecalis* (IC_50_ 18.8 μg/mL), *Listeria monocytogenes* (IC_50_ 18.8 μg/mL), *Staphylococcus aureus* (IC_50_ 9.4 μg/mL), *S. aureus*RASA (IC_50_ 9.4 μg/mL), *S. agalactiae* (IC_50_ 9.4 μg/mL), *S. epidermidis* (IC_50_ 9.4 μg/mL), *S. pyrogenes* (IC_50_ 9.4 μg/mL), Antioxidant activity (IC_50_ 0.035 mM)	[12]
*E. chinense* (Roots)	Tectorigenin (**28**)	Antimicrobial activity against with the values of (IC_50_ > 150 μg/mL) for each *Candida albicans*, *Escherichia coli*, *Klebsiella pneumoniae*, *Pseudomonas aeruginosa*, *Bacillus cereus*, *Enterococcus faecalis*, *Listeria monocytogenes*, *Staphylococcus aureus*, *S. aureus*RASA, *S. agalactiae*, *S. epidermidis* and *S. pyrogenes,* Antioxidant activity (IC_50_ 3.666 mM)	[12]
*E. chinense* (Roots)	Genistein (**29**)	Antimicrobial activity against *Candida albicans* (IC_50_ 75 μg/mL) and (IC_50_ 150 μg/mL) for each *Klebsiella pneumoniae*, *Pseudomonas aeruginosa*, *Bacillus cereus*, *Staphylococcus aureus*, *S. agalactiae* and *S. pyrogenes*	[12]
*E. chinense* (Roots)	Kaempferol (**30**)	Antimicrobial activity against with the values of (IC_50_ > 150 μg/mL) each for *Candida albicans*, *Klebsiella pneumoniae*, *Pseudomonas aeruginosa*, *Bacillus cereus*, *Staphylococcus aureus*, *S. agalactiae* and *S. pyrogenes,* Antioxidant activity (IC_50_ 0.028 mM)	[12]
*E. chinense* (Roots)	Kaempferol-7-*O*-β-d-glucopyranoside (**31**)	Antimicrobial activity against with the values of (IC_50_ > 150 μg/mL) for each *Candida albicans*, *Escherichia coli*, *Klebsiella pneumoniae*, *Pseudomonas aeruginosa*, *Bacillus cereus*, *Listeria monocytogenes*, *Staphylococcus agalactiae* and *S. epidermidis,* Antioxidant activity (IC_50_ 0.651 mM)	[12]
*E. chinense* (Roots)	Astragalin (**32**)	Antimicrobial activity against with the values of (IC_50_ 150 μg/mL) for each *Candida albicans*, *Escherichia coli*, *Klebsiella pneumoniae*, *Pseudomonas aeruginosa*, *Bacillus cereus*, *Enterococcus faecalis*, *Staphylococcus aureus*, *S. agalactiae*, *S. epidermidis* and *S. pyrogenes,* Antioxidant activity (IC_50_ 0.681 mM)	[12]
*E. suberosa* (Stem bark)	4′-methoxy licoflavanone (**36**)	The cytotoxic effects on apoptosis in human leukemia HL-60 cells and their potency to induce cancer cell death	[19]
*E. suberosa* (Stem bark)	Alpinumisoflavone (**37**)	The cytotoxic effects on apoptosis in human leukemia HL-60 cells and their potency to induce cancer cell death	[19]
*E. suberosa* (Stem bark)	Erysodine (**40**)	The anxiolytic effects in the elevated plus-maze and the light-dark transition model	[15]
*E. suberosa* (Flowers)	Erysotrine (**41**)	The anxiolytic effects in the elevated plus-maze and the light-dark transition model	[15]
*M. pendula* (Timber)	Millettilone A (**49**)	Leishmanicidal activity (IC_50_ 9.3 μg/mL)	[25]
*M. pendula* (Timber)	3*R*-Claussequinone (**50**)	Leishmanicidal activity (IC_50_ 1.2 μg/mL)	[25]
*M. pendula* (Timber)	Pendulone (**51**)	Leishmanicidal activity (IC_50_ 0.07 μg/mL)	[25]
*M. pendula* (Timber)	Secundiflorol I (**53**)	Leishmanicidal activity (IC*50* 86 μg/mL)	[25]
*M. pendula* (Timber)	3,8-Dihydroxy-9-methoxy pterocarpan (**54**)	Leishmanicidal activity (IC_50_ 2.9 μg/mL)	[25]
*M. pendula* (Timber)	3,10-Dihydroxy-7,9-dimethoxypterocarpan (**55**)	Leishmanicidal activity (IC_50_ 77 μg/mL)	[25]
*S. grandiflora* (Roots)	Isovestitol (**57**)	Antituberculosis activity against *M. tuberculosis* H37Rv (MIC 50 μg/mL)	[27]
*S. grandiflora* (Roots)	Medicarpin (**58**)	Antituberculosis activity against *M. tuberculosis* H37Rv (MIC 50 μg/mL)	[27]
*S. grandiflora* (Roots)	Sativan (**59**)	Antituberculosis activity against *M. tuberculosis* H37Rv (MIC 50 μg/mL)	[27]
*S. grandiflora* (Roots)	Betulinic acid (**60**)	Antituberculosis activity against *M. tuberculosis* H37Rv (MIC 100 μg/mL)	[27]
*T. treiquetrum* (whole plant)	Tadehaginosin (**69**)	Hypoglycemic activity in vitro by HepG2 cells	[45]
*T. treiquetrum* (whole plant)	3,4-Dihydro-4-(4′-hydroxyphenyl)-5,7-dihydroxycoumarin (**70**)	Hypoglycemic activity in vitro by HepG2 cells	[45]
*T. treiquetrum* (Aerial part)	Tadehaginosides C–J (**73**–**80**)	Antidiabetic activity	[46]
*T. treiquetrum* (Aerial part)	Tadehaginoside (**81**)	Antidiabetic activity	[46]
*A. echioides* (whole plant)	Dihydroechioidinin (**82**)	Anti-inflammatory activity with the IC_50_ of 37.6 ± 1.2 μM	[58]
*A. echioides* (whole plant)	5,7,8-Trimethoxyflavone (**98**)	Anti-inflammatory activity with the IC_50_ of 39.1 ± 1.3 μM	[58]
*J. gendarussa* (Stem and Bark)	Justiprocumin B (**111**)	Anti-HIV activity against a broad spectrum of HIV strains with IC_50_ values in the range of 15–21 nM (AZT, IC_50_ 77–95 nM), nevirapine resistant isolate HIV-1N119 with an IC_50_ value of 495 nM and AZT resistant isolate HIV-11617-1 with (IC_50_ 185 nM)	[80]
*J. gendarussa* (Stem and Bark)	Patentiflorin A (**116**)	Anti-HIV activity against a broad spectrum of HIV strains with IC_50_ values in the range of 24-37 nM (AZT, IC_50_ 77–95 nM), drug-resistant HIV-1 isolate of both the nucleotide analogue (AZT) and (nevirapine)	[82]
*P. integrifolia* (Stem bark)	10-*O*-*trans*-*p*-Methoxycinnamoyl catalpol (**119**)	Antioxidant activity with the IC_50_ value of 0.37 μM/mL in DPPH free radical scavenging assay	[101]
*P. integrifolia* (Stem bark)	4″-Hydroxy-*E*-globularinin (**125**)	Antioxidant activity with the IC_50_ value of 0.29 μM/mL in DPPH free radical scavenging assay	[101]
*P. integrifolia* (Stem bark)	Premnosidic acid (**126**)	Antioxidant activity	[101]
*P. integrifolia* (Stem bark)	Premnadimer (**127**)	Antioxidant activity	[101]
*P. integrifolia* (Stem bark)	4β-Hydroxyasarinin-1-*O*-β-glucopyranoside (**128**)	Antioxidant activity	[101]
*V. trifolia* (Aerial part)	Agnuside (**152**)	Antioxidant activity (IC_50_ 9.81 μg) in DPPH and (IC_50_ 12.90 μg) NO radical scavenging assays	[114]
*V. trifolia* (Aerial part)	Negundoside (**153**)	Antioxidant activity (IC_50_ 9.96 μg) in DPPH and (IC_50_ 16.25 μg) NO radical scavenging assays	[114]
*V. trifolia* (Aerial part)	6-*p*-Hydroxybenzoyl mussaenosidic acid (**154**)	Antioxidant activity (IC_50_ 10.31 μg) in DPPH and (IC_50_ 13.51 μg) and NO radical scavenging assays	[114]
*V. trifolia* (Leaves)	Methyl-*p*-hydroxybenzoate (**155**)	Mosquito larvicidal activity against LC_50_ values of methyl-*p*-hydroxybenzoate were 5.77 ppm against *Culex quinquefasciatus* and 4.74 ppm against *Aedes aegypti*	[115]
*A. pennata* (Leaves)	Taepeenin D (**156**)	Hedgehog/GLi-mediated transcriptional activity with IC_50_ value of 1.6 Μm, Cytotoxic against pancreatic (PANC1) cells (IC_50_ 3.2 μM) and prostate (DU145) cells (IC_50_ 3.4 μM)	[122]
*A. pennata* (Leaves)	(+)-Drim-8-ene (**157**)	Hedgehog/GLi-mediated transcriptional activity with IC_50_ value of 13.5 μM, Cytotoxic against pancratic (PANC1) cells (IC_50_ 15.1 μM) and prostate (DU145) cells (IC_50_ 23.2 μM)	[122]
*A. pennata* (Leaves)	Quercetin 3-*O*-β-d-glucopyranosyl-4-*O*-β-d-glucopyranoside (**158**)	Hedgehog/GLi-mediated transcriptional activity with IC_50_ value of 10.5 μM, Cytotoxic against pancreatic (PANC1) cells (IC_50_ 26.6 μM) and prostate (DU145) cells (IC_50_ 30.0 μM)	[122]
*C.auriculata* (Leaves)	Pseudosemiglabrin (**187**)	Hepatoprotective effects [inhibition % 37.0 ± 3.7 (*p* ˂ 0.01)] at 30 μM	[135]
*C. auriculata* (Leaves)	(2*S*)-7,4-Dihydroxy flavan (4β→8)-catechin (**188**)	Hepatoprotective effects [inhibition % 33.5 ± 2.9 (*p* < 0.01)] at 30 μM	[135]
*C. auriculata* (Leaves)	(2*S*)-7,4-Dihydroxy flavan (4β→8)-gallocatechin (**189**)	Hepatoprotective effects [inhibition % 28.2 ± 4.7 (*p* < 0.01)] at 30 μM	[135]
*C. auriculata* (Leaves)	(−)-Epigallocatechin (**192**)	Hepatoprotective effects [inhibition % 24.4 ± 3.1 (*p* < 0.01)] at 30 μM	[135]
*C. oblongifolius* (Leaves)	Nasimalun A (**193**)	Cytotoxicity toward MOLT-3 cell line with IC_50_ 26.44 μg/mL, Antibacterial activity MIC value of 50 μg/mL for *Bacilus cereus* and 100 μg/mL for *Staphylococcus aureus* and *Staphylococcus epidermidis*	[139]
*C. oblongifolius* (Stem bark)	Furanocembranoid 1 (**201**)	Cytotoxic effects against human tumor cell lines BT474 (IC_50_ 7.8 μg/mL), CHAGO (IC_50_ 7.0 μg/mL), Hep-G2 (IC_50_ 5.6 μg/mL), KATO-3 (IC_50_ 5.9 μg/mL) and SW-620s (IC_50_ 6.3 μg/mL) by MTT colorimetric method	[142]
*C. oblongifolius* (Stem bark)	Furanocembranoid 3 (**203**)	Cytotoxic effects against human tumor cell lines BT474 (IC_50_ 9.6 μg/mL), CHAGO (IC_50_ 7.1 μg/mL), Hep-G2 (IC_50_ 5.7 μg/mL), KATO-3 (IC_50_ 8.2 μg/mL) and SW-620s (IC_50_ 5.6 μg/mL) by MTT colorimetric method	[142]
*C. oblongifolius* (Stem bark)	Furanocembranoid 4 (**204**)	Cytotoxic effects against human tumor cell lines BT474 (IC_50_ 9.6 μg/mL), CHAGO (IC_50_ 9.3 μg/mL), Hep-G2 (IC_50_ 6.1 μg/mL), KATO-3 (IC_50_ 8.1 μg/mL) and SW-620s (IC_50_ 6.0 μg/mL) by MTT colorimetric method	[142]
*C. oblongifolius* (Stem bark)	(−)-ent-kuar-16-en-19-oic acid (**205**)	Inhibition of Na^+^, K^+^-ATPase activity with an IC_50_ of 2.2 × 10^−5^ M	[138]
*C. oblongifolius* (Stem bark)	Croblongifolin (**206**)	Cytotoxic effects against human tumor cell lines including HEP-G2 (IC_50_ 0.35 μM), BT474 (IC_50_ 0.12 μM), SW-620 (IC_50_ 0.47 μM) CHAGO (IC_50_ 0.24 μM) and KATO-3 (IC_50_ 0.35 μM)	[143]
*C. oblongifolius* (Stem bark)	Neocrotocembranal (**215**)	Cytotoxicity against P-388 cell culture in vitro (IC_50_ 6.48 μg/mL)	[145]
*G. pentaphylla* (Stem)	Biscarbalexine A (**227**)	Cytotoxicity against human cancer cell lines A549 (IC_50_ 56.06 μM), HepG-2 (IC_50_ 60.06 μM) and Huh-7 (IC_50_ 73.16 μM)	[156]
*G. pentaphylla* (Stem)	Glycosmisine A (**231**)	Cytotoxicity against human cancer cell lines A549 (IC_50_ 43.68 μM), HepG-2 (IC_50_ 50.30 μM) and Huh-7 (IC_50_ 30.60 μM)	[156]
*G. pentaphylla* (Stem)	Glycosmisine B (**232**)	Cytotoxicity against human cancer cell lines A549 (IC_50_ 57.10 μM), HepG-2 (IC_50_ 62.89 μM) and Huh-7 (IC_50_ 62.87 μM)	[156]

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
