# Peer review of "A Review on the Phytochemistry, Medicinal Properties and Pharmacological Activities of 15 Selected Myanmar Medicinal Plants"

_molecules, 2019, doi:10.3390/molecules24020293_

Round 1
Reviewer 1 Report
MS Title: A review on Phytochemistry, Medicinal Properties and Pharmacological Activities of 15 Selected Myanmar Medicinal Plants
Manuscript ID: molecules-411333
This is a review on the phytochemistry , medicinal properties and pharmacological activities of 15 selected Myanmar medicinal plants. Interestingly there other published review on the same topic.
Aung HT, Sein MM, Aye MM, Thu ZM. Nat Prod Commun. 2016 Mar;11(3):353-64. A Review of Traditional Medicinal Plants from Kachin State, Northern Myanmar.
Comments:
Page 1. Introduction. I suggest to mention why is important to write another review on the same topic.
Page 1. Introduction. In the end section it could be important to mention that the scientific information comes from experimental, in vitro and clinical studies in some of them.
Page 3. 2. Phytochemistry, medicinal properties and ethnopharmacology of the selected Myanmar medicinal plants. I suggest to review only those medicinal plants from which there are some published information in the scientific literature.
Pages 3-22. As the authors mentioned the scientific information is not available from following plants. I suggest to cut them.
2.1. Dalbergia culrata Grah. (DC). No reports for medicinal uses have been
found in the literature
2.4. Millettia pendula BENTH. No reports for medicinal uses have been
Tadehagi triquetrum (L.) H. Ohashi. Phytochemical constituents and biological activities of T. triquetrum have been established, however, further pharmacological studies are still missing to prove scientifically.
What about the adverse effect of these plants?
Page 31. 4. Conclusions section. I suggest to mention which plants are more likely to use in the clinical setting.
Author Response
Dear Reviewer,
Thank you very much for your helpful comments and suggestions about our manuscript, ID: molecules-411333, title "A review on Phytochemistry, Medicinal Properties and Pharmacological Activities of 15 Selected Myanmar Medicinal Plants". We have modified the manuscript accordingly. The detailed responses to the reviewers’ comments are listed below. In the new document (molecules-411333 last edition) the changes made are in color. The response to the reviewers must be compared with the PDF version of the new document in this way correspond the line numbers in the manuscript and the response that is sent.
Response to Reviewer 1 Comments
Point 1: Introduction; gave the suggestion to mention why is important to write another review on the same topic (A review of Traditional Medicinal Plants from Kachin State, Northern Myanmar)
Response 1: Thank you. It has been done in the text of introduction section (Ln 49-58).
Point 2: Introduction; In the end section it could be important to mention that the scientific information comes from experimental, in vitro and clinical studies in some of them.
Response 2: Thank you. We have rewritten some sentences as suggested by reviewer in the end section of introduction (Ln 66-73)
Point 3: Page 3; Section 2. Phytochemistry, medicinal properties and ethnopharmacology of the selected Myanmar medicinal plants. I suggest to review only those medicinal plants from which there are some published information in the scientific literature.
Response 3:Thank you. We already deleted unreliable datas that havn’t scientific evidence in section 2.
Point 4: Page 3: Section 2. As the authors mentioned the scientific information is not available from the following plants. I suggest to cut them.
Response 4: Thank you. It has been done.
Point 5: Section 2.1. Dalbergia cultrataGrah. (DC). No reports for medicinal uses have been found in the literature.
Response 5: Thank you. We was deleted the sentence that suggested by reviewer.
Point 6: Section 2.2. Millettia pendula BENTH. No reports for medicinal uses have been.
Response 6: Thank you. It has been done in the text from the Section 2.2.
Point 7: Tadehagi triquetrum (L.) H. Ohashi. Phytochemical constituents and biological activities of T. triquetrum have been established. However, further pharmacological studies are still missing to prove scientifically. What about the adverse effect of these plants?
Response 7: Thank you. We have been rewritten the details description of some sentences in the Section 2.6 of Tadehagi triquetrum(L.) H. Ohashi. (Ln 261-275)
Point 8: 4. Conclusion section. I suggest to mention which plants are more likely to use in the clinical setting.
Response 8: Thank you. In conclusion section, We strongly recommended the plants Tadehagi triquetrum andCroton oblongifolius are the most likely to use in applications of clinical setting.
Thank you for your time and consideration
With best regards,
Chabaco Patricio Armijos Riofrío, Ph.D.
Departamento de Química
Universidad Técnica Particular de Loja.
Tel: +593 7 3701444, Loja - Ecuador
E-mail address: cparmijos@utpl.edu.ec
Reviewer 2 Report
The authors have conducted a review of the Phytochemistry, Medicinal Properties and Pharmacological Activities of 15 Selected Myanmar Medicinal Plants. While this provides potentially important information, it is unclear why this group of plants was selected given the large numbers of possible medicinal plants found in the region [ref. 2: Soe et al, 2004] Further, a common description for each plant is that they are used in treatment of a range of disorders, but these are very generic [e.g. treatment of diabetes, inflammations, swellings, bronchitis, dyspepsia, liver disorders, piles, constipation, and fever] and give little no indications as to the nature of the bioactive component, what exactly they are used to treat and how, dosage or treatment regime. Indeed, it is unclear how much of the information is based on folklore usage or valid medical experience. The authors need to be clear on this and give more examples of where products from these plants have successfully been used in a specific disease/clinical setting.
The possible bioactive factors covered in the text tend to be flavonoids, polyphenols or alkaloids isolated in alcoholic extracts from various plant products. While much of the published work on medicinal plants seems to involve these groups of bioactive factors there are an array of other bioactive factors in plants that may make as much a contribution to the medical or health-promoting properties under study. I did note passing mentions to essential oils (ln 377) and an aqueous extract of leaves (ln 409) but little details. If the data presented is all that is available for the 15 plants, it needs to be specified in the text that other bioactive factors may also contribute to the actions of the plant extracts. This is of relevance because it is difficult to match the diverse array of supposed health-promoting effects with the limited number of compounds identified. Also, most findings in vivo are with extracts of an ill-defined composition.
The emphasis throughout is given to the medicinal and health-promoting effects of the plant products. However, some of the plants also express toxic components and indeed some of the factors identified in the text can have deleterious effects at high intakes. A discussion on potential adverse effects of bioactive factors from the plants is needed.
Ln 55-56 Is this reputed or known? Is there a suitable reference?
Ln 65-78 Which plant parts contain specified compounds? Are these the only bioactive compounds in the plant?
Ln 124 Need to give a reference for this statement.
Ln 162-164 What parts of the plant are used and in what form? Is statement folklore-based?
Ln 183-192 As for Ln 162-164.
Ln 239-249 As for Ln 162-164.
Ln 251-256 As for Ln 162-164.
Ln 300, 324, 338 Which plant parts and in which form?
Ln 336-337 What parts of the plant are used and in what form? Is statement folklore-based or scientifically validated?
Ln 377-378 First mention of essential oils. Need more description of findings.
Ln 409-415 & 486 Mention of aqueous extract and of essential oils. Need more information as these fractions differ from most others described in the paper.
Ln 583 ‘cytotoxic’. Not sure this is an appropriate word. ‘bioactive’?
Ln 585-588 Need to discuss overall findings from table 2.
‘Currently, there is a shift from a single to a multiple target approach in drug discovery’. What does this mean?
Ln 590 This is a very useful table, but the first page has miscued in pdf formatting.
Author Response
Dear Reviewer,
Thank you very much for your helpful comments and suggestions about our manuscript, ID: molecules-411333, title "A review on Phytochemistry, Medicinal Properties and Pharmacological Activities of 15 Selected Myanmar Medicinal Plants". We have modified the manuscript accordingly. The detailed responses to the reviewers’ comments are listed below. In the new document (molecules-411333 last edition) the changes made are in color. The response to the reviewers must be compared with the PDF version of the new document in this way correspond the line numbers in the manuscript and the response that is sent.
Response to Reviewer 2 Comments
Point 1: Ln 55-56. Is this reputed or known? Is there a suitable reference?
Response 1: Thank you. Ln 55-56 in submitted manuscript. This is reputed. It hasn’t suitable reference. Therefore it has been deleted.
Point 2: Ln 65-78. Which plant parts contain specified compounds? Are these the only bioactive compounds in the plant?
Response 2: Thank you. (Now Ln 80-91, after editing manuscript), this plant of Family (Leguminosae) is known to contain isoflavonoids and neoflavonoids. These are the only bioactive compounds in the plant reported but also another type of compounds contain in it.
Point 3: Ln 124. Need to give a reference for this statement.
Response 3: Thank you.We have added the reference in the text for this statement (Ln 138-139).
Point 4: Ln 162-164. What parts of the plant are used and in what form? Is statement folklore based?
Response 4: Thank you. It has been corrected. This statement is base on folk medicine. (Ln 179-181).
Point 5: Ln 183-192. What parts of the plant are used and in what form? Is statement folklore-based?
Response 5: Thank you. We have rewritten these sentences in the text of the section 2.5 (Ln 201-203).
Point 6: Ln 239-249. What parts of the plant are used and in what form? Is statement folklore-based?
Response 6: Thank you, we have rewritten this part in the section 2.6 of the manuscript. (Ln 254-268).
Point 7: Ln 251-256. What parts of the plant are used and in what form? Is statement folklore-based?
Response 7: Thank you. It has been edited this statement in the text of manuscript (Ln 276-278).
Point 8: Ln 300, 324, 338 Which plant parts and in which form?
Response 8: Thank you. Ln 300 is already corrected in manuscript. (Ln 319-322 in update version). Ln 324 has been done (Ln 343-345 in new version). Ln 338 could be seen in Ln 357-364) of new manuscript.
Point 9: Ln 336-337. What parts of the plant are used and in what form? Is statement folklore-based or scientifically validated?
Response 9: Thank you. This statement is not validated scientifically till now. That’s why we have been cut from the section 2.9 of manuscript.
Point 10: Ln 377-378. First mention of essential oils. Need more description of findings.
Response 10:Thank you. We have been corrected in manuscript (Ln 398-404).
Point 11: Ln 409-415 & 486. Mention of aqueous extract and of essential oils. Need more information as these fractions differ from most others described in the paper.
Response 11: Thankyou.Aqueous extract of the leaves of Vitex trifolia, from species that are in Myanmar was reported with hepatoprotective activity against acetaminophen-induced hepatotoxicity in albino rat models. This statement was already deleted because that data is not reliable. There has no identification of essential oils specifically in reference paper. Ln 486 has been corrected. (Please see Ln 512-514 in new manuscript).
Point 12: Ln 583. ‘cytotoxic’. Not sure this is an appropriate word. ‘bioactive’?
Response 12:Thank you. It has been corrected from cytotoxic to bioactive. (Ln 611)
Point 13: Ln 585-588. Need to discuss overall findings from table 2.
Response 13: Thank you. Literature reviews on biological/pharmacological activities of chemical constituents isolated from selected plants are details described in each section. Published biological and pharmacological applications of isolated compounds from these plants are condensed in Table 2.
Point 14: ‘Currently, there is a shift from a single to a multiple target approach in drug discovery’. What does it mean?
Response 14: Thank you. This statement is not concern in the conclusion section of our review. So it has been deleted.
Point 15: Ln 590 formatting. This is a very useful table. But the first page has miscued in pdf.
Response 15: Thank you. It has been done.
Thank you for your time and consideration.
With best regards,
Chabaco Patricio Armijos Riofrío, Ph.D.
Departamento de Química
Universidad Técnica Particular de Loja.
Tel: +593 7 3701444, Loja - Ecuador
E-mail address: cparmijos@utpl.edu.ec
Round 2
Reviewer 1 Report
Page 22 first paragraph, lines 611-612 are similar than those lines (617-619) in the first paragraph of the conclusion section.
Page 22 Conclusion section. It is too long and redundant. I suggest to rewrite it
Author Response
Dear Reviewer,
Thank you very much for your helpful comments and suggestions about our manuscript, ID: molecules-411333, title "A review on Phytochemistry, Medicinal Properties and Pharmacological Activities of 15 Selected Myanmar Medicinal Plants". We have modified the manuscript accordingly. The detailed responses to the reviewers’ comments are listed below.
Point 1: Page 22 first paragraph, lines 611-612 are similar than those lines 617-619 in the first paragraph of the conclusion section.
Response 1: Thank you. It has been done in conclusion section, page 22. We already deleted lines 617-619 in the first paragraph of conclusion section.
Point 2: Page 22 conclusion section. It is too long and reductant. I suggest to rewrite it.
Response 2: Thank you. We already reduced the text in the conclusion section of page 22. It has been rewritten.
Thank you for your time and consideration.
With best regards,
Chabaco Patricio Armijos Riofrío, Ph.D.
Departamento de Química
Universidad Técnica Particular de Loja.
Tel: +593 7 3701444, Loja - Ecuador
E-mail address: cparmijos@utpl.edu.ec
Reviewer 2 Report
The authors have dealt adequately with the queries raised in the review.
Author Response
Dear Reviewer,
Thank you very much for your helpful comments and suggestions about our manuscript, ID: molecules-411333, title "A review on Phytochemistry, Medicinal Properties and Pharmacological Activities of 15 Selected Myanmar Medicinal Plants". We have modified the manuscript accordingly. The detailed responses to the reviewers’ comments are listed below.
Comments: The authors have dealt adequately with the queries raised in the review.
Response:Thank you so much for your comments.
Thank you for your time and consideration.
With best regards,
Chabaco Patricio Armijos Riofrío, Ph.D.
Chemestry Department
Universidad Técnica Particular de Loja.
Tel: +593 7 3701444, Loja - Ecuador
E-mail address: cparmijos@utpl.edu.ec